# Probabilistic Differential Dynamic Programming

**Yunpeng Pan** and **Evangelos A. Theodorou**
Daniel Guggenheim School of Aerospace Engineering
Institute for Robotics and Intelligent Machines
Georgia Institute of Technology
Atlanta, GA 30332
`ypan37@gatech.edu, evangelos.theodorou@ae.gatech.edu`

## Abstract

We present a data-driven, probabilistic trajectory optimization framework for systems with unknown dynamics, called Probabilistic Differential Dynamic Programming (PDDP). PDDP takes into account uncertainty explicitly for dynamics models using Gaussian processes (GPs). Based on the second-order local approximation of the value function, PDDP performs Dynamic Programming around a nominal trajectory in Gaussian belief spaces. Different from typical gradient-based policy search methods, PDDP does not require a policy parameterization and learns a locally optimal, time-varying control policy. We demonstrate the effectiveness and efficiency of the proposed algorithm using two nontrivial tasks. Compared with the classical DDP and a state-of-the-art GP-based policy search method, PDDP offers a superior combination of data-efficiency, learning speed, and applicability.

## 1 Introduction

Differential Dynamic Programming (DDP) is a powerful trajectory optimization approach. Originally introduced in [1], DDP generates locally optimal feedforward and feedback control policies along with an optimal state trajectory. Compared with global optimal control approaches, the local optimal DDP shows superior computational efficiency and scalability to high-dimensional problems. In the last decade, variations of DDP have been proposed in both control and machine learning communities [2][3][4][5][6]. Recently, DDP was applied for high-dimensional policy search which achieved promising results in challenging control tasks [7].

DDP is derived based on linear approximations of the nonlinear dynamics along state and control trajectories, therefore it relies on accurate and explicit dynamics models. However, modeling a dynamical system is in general a challenging task and model uncertainty is one of the principal limitations of model-based methods. Various parametric and semi-parametric approaches have been developed to address these issues, such as minimax DDP using Receptive Field Weighted Regression (RFWR) by Morimoto and Atkeson [8], and DDP using expert-demonstrated trajectories by Abbeel et al. [9]. Motivated by the complexity of the relationships between states, controls and observations in autonomous systems, in this work we take a Bayesian non-parametric approach using Gaussian Processes (GPs).

Over last few years, GP-based control and Reinforcement Learning (RL) algorithms have increasingly drawn more attention in control theory and machine learning communities. For instance, the works by Rasmussen et al.[10], Nguyen-Tuong et al.[11], Deisenroth et al.[12][13][14] and Hemakumara et al.[15] have demonstrated the remarkable applicability of GP-based control and RL methods in robotics. In particular, a recently proposed GP-based policy search framework called PILCO, developed by Deisenroth and Rasmussen [13] (an improved version has been developed by Deisenroth, Fox and Rasmussen [14]) has achieved unprecedented performances in terms of data-

efficiency and policy learning speed. PILCO as well as most gradient-based policy search algorithms require iterative methods (e.g.,CG or BFGS) for solving non-convex optimization to obtain optimal policies.

The proposed approach does not require a policy parameterization. Instead PDDP finds a linear, time varying control policy based on Bayesian non-parametric representation of the dynamics and outperforms PILCO in terms of control learning speed while maintaining a comparable data-efficiency.

## 2 Proposed Approach

The proposed PDDP framework consists of 1) a Bayesian non-parametric representation of the unknown dynamics; 2) local approximations of the dynamics and value functions; 3) locally optimal controller learning.

### 2.1 Problem formulation

We consider a general unknown stochastic system described by the following differential equation

$$\mathrm{d}\mathbf{x} = \mathbf{f}(\mathbf{x}, \mathbf{u})\mathrm{d}t + \mathbf{C}(\mathbf{x}, \mathbf{u})\mathrm{d}\omega, \qquad \mathbf{x}(t_0) = \mathbf{x}_0, \qquad \mathrm{d}\omega \sim \mathcal{N}(0, \boldsymbol{\Sigma}_\omega), \tag{1}$$

where $\mathbf{x} \in \mathbb{R}^n$ is the state, $\mathbf{u} \in \mathbb{R}^m$ is the control, $t$ is time and $\omega \in \mathbb{R}^p$ is standard Brownian motion noise. The trajectory optimization problem is defined as finding a sequence of state and controls that minimize the expected cost

$$J^\pi(\mathbf{x}(t_0)) = \mathbb{E}\left[ h\Big(\mathbf{x}(T)\Big) + \int_{t_0}^{T} \mathcal{L}\Big(\mathbf{x}(t), \pi(\mathbf{x}(t)), t\Big)\mathrm{d}t \right], \tag{2}$$

where $h(\mathbf{x}(T))$ is the terminal cost, $\mathcal{L}(\mathbf{x}(t), \pi(\mathbf{x}(t)), t)$ is the instantaneous cost rate, $\mathbf{u}(t) = \pi(\mathbf{x}(t))$ is the control policy. The cost $J^\pi(\mathbf{x}(t_0))$ is defined as the expectation of the total cost accumulated from $t_0$ to $T$. For the rest of our analysis, we denote $\mathbf{x}_k = \mathbf{x}(t_k)$ in discrete-time where $k = 0, 1, ..., H$ is the time step, we use this subscript rule for other variables as well.

### 2.2 Probabilistic dynamics model learning

The continuous functional mapping from state-control pair $\tilde{\mathbf{x}} = (\mathbf{x}, \mathbf{u}) \in \mathbb{R}^{n+m}$ to state transition $\mathrm{d}\mathbf{x}$ can be viewed as an inference with the goal of inferring $\mathrm{d}\mathbf{x}$ given $\tilde{\mathbf{x}}$. We view this inference as a nonlinear regression problem. In this subsection, we introduce the Gaussian processes (GP) approach to learning the dynamics model in (1). A GP is defined as a collection of random variables, any finite number subset of which have a joint Gaussian distribution. Given a sequence of state-control pairs $\tilde{\mathbf{X}} = \{(\mathbf{x}_0, \mathbf{u}_0), \dots (\mathbf{x}_H, \mathbf{u}_H)\}$, and the corresponding state transition $\mathrm{d}\mathbf{X} = \{\mathrm{d}\mathbf{x}_0, \dots, \mathrm{d}\mathbf{x}_H\}$, a GP is completely defined by a mean function and a covariance function. The joint distribution of the observed output and the output corresponding to a given test state-control pair $\tilde{\mathbf{x}}^* = (\mathbf{x}^*, \mathbf{u}^*)$ can be written as $\mathrm{p}\left( \begin{smallmatrix} \mathrm{d}\mathbf{X} \\ \mathrm{d}\mathbf{x}^* \end{smallmatrix} \right) \sim \mathcal{N}\left(0, \left[ \begin{smallmatrix} \mathbf{K}(\tilde{\mathbf{X}}, \tilde{\mathbf{X}}) + \sigma_n\mathbf{I} & \mathbf{K}(\tilde{\mathbf{X}}, \tilde{\mathbf{x}}^*) \\ \mathbf{K}(\tilde{\mathbf{x}}^*, \tilde{\mathbf{X}}) & \mathbf{K}(\tilde{\mathbf{x}}^*, \tilde{\mathbf{x}}^*) \end{smallmatrix} \right] \right)$. The covariance of this multivariate Gaussian distribution is defined via a kernel matrix $\mathbf{K}(\mathbf{x}_i, \mathbf{x}_j)$. In particular, in this paper we consider the Gaussian kernel $\mathbf{K}(\mathbf{x}_i, \mathbf{x}_j) = \sigma_s^2 \exp(-\frac{1}{2}(\mathbf{x}_i - \mathbf{x}_j)^\mathrm{T}\mathbf{W}(\mathbf{x}_i - \mathbf{x}_j)) + \sigma_n^2$, with $\sigma_s, \sigma_n, \mathbf{W}$ the hyper-parameters. The kernel function can be interpreted as a similarity measure of random variables. More specifically, if the training pairs $\tilde{\mathbf{X}}_i$ and $\tilde{\mathbf{X}}_j$ are close to each other in the kernel space, their outputs $\mathrm{d}\mathbf{x}_i$ and $\mathrm{d}\mathbf{x}_j$ are highly correlated. The posterior distribution, which is also a Gaussian, can be obtained by constraining the joint distribution to contain the output $\mathrm{d}\mathbf{x}^*$ that is consistent with the observations. Assuming independent outputs (no correlation between each output dimension) and given a test input $\tilde{\mathbf{x}}_k = (\mathbf{x}_k, \mathbf{u}_k)$ at time step $k$, the one-step predictive mean and variance of the state transition are specified as

$$\mathbb{E}_\mathbf{f}[\mathrm{d}\mathbf{x}_k] = \mathbf{K}(\tilde{\mathbf{x}}_k, \tilde{\mathbf{X}})(\mathbf{K}(\tilde{\mathbf{X}}, \tilde{\mathbf{X}}) + \sigma_n\mathbf{I})^{-1}\mathrm{d}\mathbf{X}, \tag{3}$$

$$\mathbb{VAR}_\mathbf{f}[\mathrm{d}\mathbf{x}_k] = \mathbf{K}(\tilde{\mathbf{x}}_k, \tilde{\mathbf{x}}_k) - \mathbf{K}(\tilde{\mathbf{x}}_k, \tilde{\mathbf{X}})(\mathbf{K}(\tilde{\mathbf{X}}, \tilde{\mathbf{X}}) + \sigma_n\mathbf{I})^{-1}\mathbf{K}(\tilde{\mathbf{X}}, \tilde{\mathbf{x}}_k).$$

The state distribution at $k = 1$ is $\mathrm{p}(\mathbf{x}_1) \sim \mathcal{N}(\boldsymbol{\mu}_1, \boldsymbol{\Sigma}_1)$ where the state mean and variance are $\boldsymbol{\mu}_1 = \mathbf{x}_0 + \mathbb{E}_\mathbf{f}[\mathrm{d}\mathbf{x}_0], \boldsymbol{\Sigma}_1 = \mathbb{VAR}_\mathbf{f}[\mathrm{d}\mathbf{x}_0]$. When propagating the GP-based dynamics over a trajectory of time horizon $H$, the input state-control pair $\tilde{\mathbf{x}}_k$ becomes uncertain with a Gaussian distribution

(initially $\tilde{\mathbf{x}}_0$ is deterministic). Here we define the joint distribution over state-control pair at $k$ as $\mathrm{p}(\tilde{\mathbf{x}}_k) = \mathrm{p}(\mathbf{x}_k, \mathbf{u}_k) \sim \mathcal{N}(\tilde{\boldsymbol{\mu}}_k, \tilde{\boldsymbol{\Sigma}}_k)$. Thus the distribution over state transition becomes $\mathrm{p}(\mathrm{d}\mathbf{x}_k) = \int \mathrm{p}(\mathbf{f}(\tilde{\mathbf{x}}_k)|\tilde{\mathbf{x}}_k)\mathrm{p}(\tilde{\mathbf{x}}_k)\mathrm{d}\tilde{\mathbf{x}}_k$. Generally, this predictive distribution cannot be computed analytically because the nonlinear mapping of an input Gaussian distribution lead to a non-Gaussian predictive distribution. However, the predictive distribution can be approximated by a Gaussian $\mathrm{p}(\mathrm{d}\mathbf{x}_k) \sim \mathcal{N}(\mathrm{d}\boldsymbol{\mu}_k, \mathrm{d}\boldsymbol{\Sigma}_k)$ [16]. Thus the state distribution at $k+1$ is also a Gaussian $\mathcal{N}(\boldsymbol{\mu}_{k+1}, \boldsymbol{\Sigma}_{k+1})$ [14]

$$\boldsymbol{\mu}_{k+1} = \boldsymbol{\mu}_k + \mathrm{d}\boldsymbol{\mu}_k, \qquad \boldsymbol{\Sigma}_{k+1} = \boldsymbol{\Sigma}_k + \mathrm{d}\boldsymbol{\Sigma}_k + \mathbb{COV}_{\mathbf{f},\tilde{\mathbf{x}}_k}[\mathbf{x}_k, \mathrm{d}\mathbf{x}_k] + \mathbb{COV}_{\mathbf{f},\tilde{\mathbf{x}}_k}[\mathrm{d}\mathbf{x}_k, \mathbf{x}_k]. \quad (4)$$

Given an input joint distribution $\mathcal{N}(\tilde{\boldsymbol{\mu}}_k, \tilde{\boldsymbol{\Sigma}}_k)$, we employ the moment matching approach [16][14] to compute the posterior GP. The predictive mean $\mathrm{d}\boldsymbol{\mu}_k$ is evaluated as

$$\mathrm{d}\boldsymbol{\mu}_k = \mathbb{E}_{\tilde{\mathbf{x}}_k}\big[\mathbb{E}_{\mathbf{f}}[\mathrm{d}\mathbf{x}_k]\big] = \int \mathbb{E}_{\mathbf{f}}[\mathrm{d}\mathbf{x}_k]\mathcal{N}(\tilde{\boldsymbol{\mu}}_k, \tilde{\boldsymbol{\Sigma}}_k)\mathrm{d}\tilde{\mathbf{x}}_k.$$

Next, we compute the predictive covariance matrix

$$\mathrm{d}\boldsymbol{\Sigma}_k = \begin{bmatrix} \mathbb{VAR}_{\mathbf{f},\tilde{\mathbf{x}}_k}[\mathrm{d}\mathbf{x}_{k_1}] & \cdots & \mathbb{COV}_{\mathbf{f},\tilde{\mathbf{x}}_k}[\mathrm{d}\mathbf{x}_{k_n}, \mathrm{d}\mathbf{x}_{k_1}] \\ \vdots & \ddots & \vdots \\ \mathbb{COV}_{\mathbf{f},\tilde{\mathbf{x}}_k}[\mathrm{d}\mathbf{x}_{k_1}, \mathrm{d}\mathbf{x}_{k_n}] & \cdots & \mathbb{VAR}_{\mathbf{f},\tilde{\mathbf{x}}_k}[\mathrm{d}\mathbf{x}_{k_n}] \end{bmatrix},$$

where the variance term on the diagonal for output dimension $i$ is obtained as

$$\mathbb{VAR}_{\mathbf{f},\tilde{\mathbf{x}}_k}[\mathrm{d}\mathbf{x}_{k_i}] = \mathbb{E}_{\tilde{\mathbf{x}}_k}\big[\mathbb{VAR}_{\mathbf{f}}[\mathrm{d}\mathbf{x}_{k_i}]\big] + \mathbb{E}_{\tilde{\mathbf{x}}_k}\big[\mathbb{E}_{\mathbf{f}}[\mathrm{d}\mathbf{x}_{k_i}]^2\big] - \mathbb{E}_{\tilde{\mathbf{x}}_k}\big[\mathbb{E}_{\mathbf{f}}[\mathrm{d}\mathbf{x}_{k_i}]\big]^2, \quad (5)$$

and the off-diagonal covariance term for output dimension $i, j$ is given by the expression

$$\mathbb{COV}_{\mathbf{f},\tilde{\mathbf{x}}_k}[\mathrm{d}\mathbf{x}_{k_i}, \mathrm{d}\mathbf{x}_{k_j}] = \mathbb{E}_{\tilde{\mathbf{x}}_k}\big[\mathbb{E}_{\mathbf{f}}[\mathrm{d}\mathbf{x}_{k_i}]\mathbb{E}_{\mathbf{f}}[\mathrm{d}\mathbf{x}_{k_j}]\big] - \mathbb{E}_{\tilde{\mathbf{x}}_k}[\mathbb{E}_{\mathbf{f}}[\mathrm{d}\mathbf{x}_{k_i}]]\mathbb{E}_{\tilde{\mathbf{x}}_k}[\mathbb{E}_{\mathbf{f}}[\mathrm{d}\mathbf{x}_{k_j}]]. \quad (6)$$

The input-output cross-covariance is formulated as

$$\mathbb{COV}_{\mathbf{f},\tilde{\mathbf{x}}_k}[\tilde{\mathbf{x}}_k, \mathrm{d}\mathbf{x}_k] = \mathbb{E}_{\tilde{\mathbf{x}}_k}\big[\tilde{\mathbf{x}}_k\mathbb{E}_{\mathbf{f}}[\mathrm{d}\mathbf{x}_k]^{\mathrm{T}}\big] - \mathbb{E}_{\tilde{\mathbf{x}}_k}[\tilde{\mathbf{x}}_k]\mathbb{E}_{\mathbf{f},\tilde{\mathbf{x}}_k}[\mathrm{d}\mathbf{x}_k]^{\mathrm{T}}. \quad (7)$$

$\mathbb{COV}_{\mathbf{f},\tilde{\mathbf{x}}_k}[\mathbf{x}_k, \mathrm{d}\mathbf{x}_k]$ can be easily obtained as a sub-matrix of (7). The kernel or hyper-parameters $\Theta = (\sigma_n, \sigma_s, \mathbf{W})$ can be learned by maximizing the log-likelihood of the training outputs given the inputs

$$\Theta^* = \underset{\Theta}{\mathrm{argmax}}\left\{ \log\left(\mathrm{p}\big(\mathrm{d}\mathbf{X}|\tilde{\mathbf{X}}, \Theta\big)\right) \right\}. \quad (8)$$

This optimization problem can be solved using numerical methods such as conjugate gradient [17].

## 2.3 Local dynamics model

In DDP-related algorithms, a local model along a nominal trajectory $(\bar{\mathbf{x}}_k, \bar{\mathbf{u}}_k)$, is created based on: i) a first or second-order linear approximation of the dynamics model; ii) a second-order local approximation of the value function. In our proposed PDDP framework, we will create a local model along a trajectory of state distribution-control pair $(\mathrm{p}(\bar{\mathbf{x}}_k), \bar{\mathbf{u}}_k)$. In order to incorporate uncertainty explicitly in the local model, we introduce the Gaussian belief augmented state vector $\mathbf{z}_k^x = [\boldsymbol{\mu}_k \ \mathrm{vec}(\boldsymbol{\Sigma}_k)]^{\mathrm{T}} \in \mathbb{R}^{n+n \times n}$ where $\mathrm{vec}(\boldsymbol{\Sigma}_k)$ is the vectorization of $\boldsymbol{\Sigma}_k$. Now we create a local linear model of the dynamics. Based on eq.(4), the dynamics model with the augmented state is

$$\mathbf{z}_{k+1}^x = \mathcal{F}(\mathbf{z}_k^x, \mathbf{u}_k). \quad (9)$$

Define the control and state variations $\delta\mathbf{z}_k^x = \mathbf{z}_k^x - \bar{\mathbf{z}}_k^x$ and $\delta\mathbf{u}_k = \mathbf{u}_k - \bar{\mathbf{u}}_k$. In this work we consider the first-order expansion of the dynamics. More precisely we have

$$\delta\mathbf{z}_{k+1}^x = \mathcal{F}_k^x \delta\mathbf{z}_k^x + \mathcal{F}_k^u \delta\mathbf{u}_k, \quad (10)$$

where the Jacobian matrices $\mathcal{F}_k^x$ and $\mathcal{F}_k^u$ are specified as

$$\mathcal{F}_k^x = \nabla_{\mathbf{x}_k}\mathcal{F} = \begin{bmatrix} \frac{\partial\boldsymbol{\mu}_{k+1}}{\partial\boldsymbol{\mu}_k} & \frac{\partial\boldsymbol{\mu}_{k+1}}{\partial\boldsymbol{\Sigma}_k} \\ \frac{\partial\boldsymbol{\Sigma}_{k+1}}{\partial\boldsymbol{\mu}_k} & \frac{\partial\boldsymbol{\Sigma}_{k+1}}{\partial\boldsymbol{\Sigma}_k} \end{bmatrix} \in \mathbb{R}^{(n+n^2) \times (n+n^2)},$$

$$\mathcal{F}_k^u = \nabla_{\mathbf{u}_k}\mathcal{F} = \begin{bmatrix} \frac{\partial\boldsymbol{\mu}_{k+1}}{\partial\mathbf{u}_k} \\ \frac{\partial\boldsymbol{\Sigma}_{k+1}}{\partial\mathbf{u}_k} \end{bmatrix} \in \mathbb{R}^{(n+n^2) \times m}.$$

$$(11)$$

The partial derivatives $\frac{\partial\boldsymbol{\mu}_{k+1}}{\partial\boldsymbol{\mu}_k}, \frac{\partial\boldsymbol{\mu}_{k+1}}{\partial\boldsymbol{\Sigma}_k}, \frac{\partial\boldsymbol{\Sigma}_{k+1}}{\partial\boldsymbol{\mu}_k}, \frac{\partial\boldsymbol{\Sigma}_{k+1}}{\partial\boldsymbol{\Sigma}_k}, \frac{\partial\boldsymbol{\mu}_{k+1}}{\partial\mathbf{u}_k}, \frac{\partial\boldsymbol{\Sigma}_{k+1}}{\partial\mathbf{u}_k}$ can be computed analytically. Their forms are provided in the supplementary document of this work. For numerical implementation, the dimension of the augmented state can be reduced by eliminating the redundancy of $\boldsymbol{\Sigma}_k$ and the principle square root of $\boldsymbol{\Sigma}_k$ may be used for numerical robustness [6].

## 2.4 Cost function

In the classical DDP and many optimal control problems, the following quadratic cost function is used

$$\mathcal{L}(\mathbf{x}_k, \mathbf{u}_k) = (\mathbf{x}_k - \mathbf{x}_k^{goal})^\mathrm{T}\mathbf{Q}(\mathbf{x}_k - \mathbf{x}_k^{goal}) + \mathbf{u}_k^\mathrm{T}\mathbf{R}\mathbf{u}_k, \tag{12}$$

where $\mathbf{x}_k^{goal}$ is the target state. Given the distribution $\mathrm{p}(\mathbf{x}_k) \sim \mathcal{N}(\boldsymbol{\mu}_k, \boldsymbol{\Sigma}_k)$, the expectation of original quadratic cost function is formulated as

$$\mathbb{E}_{\mathbf{x}_k}\Big[\mathcal{L}(\mathbf{x}_k, \mathbf{u}_k)\Big] = \mathrm{tr}(\mathbf{Q}\boldsymbol{\Sigma}_k) + (\boldsymbol{\mu}_k - \mathbf{x}_k^{goal})^\mathrm{T}\mathbf{Q}(\boldsymbol{\mu}_k - \mathbf{x}_k^{goal}) + \mathbf{u}_k^\mathrm{T}\mathbf{R}\mathbf{u}_k. \tag{13}$$

In PDDP, we use the cost function $\mathcal{L}(\mathbf{z}_k^x, \mathbf{u}_k) = \mathbb{E}_{\mathbf{x}_k}[\mathcal{L}(\mathbf{x}_k, \mathbf{u}_k)]$. The analytic expressions of partial derivatives $\frac{\partial}{\partial \mathbf{z}_k^x}\mathcal{L}(\mathbf{z}_k^x, \mathbf{u}_k)$ and $\frac{\partial}{\partial \mathbf{u}_k}\mathcal{L}(\mathbf{z}_k^x, \mathbf{u}_k)$ can be easily obtained. The cost function (13) scales linearly with the state covariance, therefore the exploration strategy of PDDP is balanced between the distance from the target and the variance of the state. This strategy fits well with DDP-related frameworks that rely on local approximations of the dynamics. A locally optimal controller obtained from high-risk explorations in uncertain regions might be highly undesirable.

## 2.5 Control policy

The Bellman equation for the value function in discrete-time is specified as follows

$$V(\mathbf{z}_k^x, k) = \min_{\mathbf{u}_k} \mathbb{E}\left[ \underbrace{\mathcal{L}(\mathbf{z}_k^x, \mathbf{u}_k) + V\Big(\mathcal{F}(\mathbf{z}_k^x, \mathbf{u}_k), k+1\Big)}_{\mathbf{Q}(\mathbf{z}_k^x, \mathbf{u}_k)} |\mathbf{x}_k \right]. \tag{14}$$

We create a quadratic local model of the value function by expanding the $\mathbf{Q}$-function up to the second order

$$\mathbf{Q}_k(\mathbf{z}_k^x + \delta\mathbf{z}_k^x, \mathbf{u}_k + \delta\mathbf{u}_k) \approx \mathbf{Q}_k^0 + \mathbf{Q}_k^x \delta\mathbf{z}_k^x + \mathbf{Q}_k^u \delta\mathbf{u}_k + \frac{1}{2}\begin{bmatrix} \delta\mathbf{z}_k^x \\ \delta\mathbf{u}_k \end{bmatrix}^\mathrm{T} \begin{bmatrix} \mathbf{Q}_k^{xx} & \mathbf{Q}_k^{xu} \\ \mathbf{Q}_k^{ux} & \mathbf{Q}_k^{uu} \end{bmatrix} \begin{bmatrix} \delta\mathbf{z}_k^x \\ \delta\mathbf{u}_k \end{bmatrix}, \tag{15}$$

where the superscripts of the $\mathbf{Q}$-function indicate derivatives. For instance, $\mathbf{Q}_k^x = \nabla_x\mathbf{Q}_k(\mathbf{z}_k^x, \mathbf{u}_k)$. For the rest of the paper, we will use this superscript rule for $\mathcal{L}$ and $V$ as well. To find the optimal control policy, we compute the local variations in control $\delta\hat{\mathbf{u}}_k$ that maximize the $\mathbf{Q}$-function

$$\delta\hat{\mathbf{u}}_k = \arg\max_{\mathbf{u}_k}\Big[\mathbf{Q}_k(\mathbf{z}_k^x + \delta\mathbf{z}_k^x, \mathbf{u}_k + \delta\mathbf{u}_k)\Big] = \underbrace{-(\mathbf{Q}_k^{uu})^{-1}\mathbf{Q}_k^u}_{\mathbf{I}_k} \underbrace{-(\mathbf{Q}_k^{uu})^{-1}\mathbf{Q}_k^{ux}}_{\mathbf{L}_k}\delta\mathbf{z}_k^x = \mathbf{I}_k + \mathbf{L}_k\delta\mathbf{z}_k^x.$$
$$\tag{16}$$

The optimal control can be found as $\hat{\mathbf{u}}_k = \bar{\mathbf{u}}_k + \delta\hat{\mathbf{u}}_k$. The quadratic expansion of the value function is backward propagated based on the equations that follow

$$\mathbf{Q}_k^x = \mathcal{L}_k^x + V_k^x\mathcal{F}_k^x, \quad \mathbf{Q}_k^u = \mathcal{L}_k^u + V_k^x\mathcal{F}_k^u,$$
$$\mathbf{Q}_k^{xx} = \mathcal{L}_k^{xx} + (\mathcal{F}_k^x)^\mathrm{T}V_k^{xx}\mathcal{F}_k^x, \quad \mathbf{Q}_k^{ux} = \mathcal{L}_k^{ux} + (\mathcal{F}_k^u)^\mathrm{T}V_k^{xx}\mathcal{F}_k^x, \quad \mathbf{Q}_k^{uu} = \mathcal{L}_k^{uu} + (\mathcal{F}_k^u)^\mathrm{T}V_k^{xx}\mathcal{F}_k^u,$$
$$V_{k-1} = V_k + \mathbf{Q}_k^u\mathbf{I}_k, \quad V_{k-1}^x = \mathbf{Q}_k^x + \mathbf{Q}_k^u\mathbf{L}_k, \quad V_{k-1}^{xx} = \mathbf{Q}_k^{xx} + \mathbf{Q}_k^{xu}\mathbf{L}_k. \tag{17}$$

The second-order local approximation of the value function is propagated backward in time iteratively. We use the learned controller to generate a locally optimal trajectory by propagating the dynamics forward in time. The control policy is a linear function of the augmented state $\mathbf{z}_k^x$, therefore the controller is deterministic. The state propagations have been discussed in Sec. 2.2.

## 2.6 Summary of algorithm

The proposed algorithm can be summarized in **Algorithm 1**. The algorithm consists of 8 modules. In *Model learning* (Step 1-2) we sample trajectories from the original physical system in order to collect training data and learn a probabilistic model. In *Local approximation* (Step 4) we obtain a local linear approximation (10) of the learned probabilistic model along a nominal trajectory by computing Jacobian matrices (11). In *Controller learning* (Step 5) we compute a local optimal control sequence (16) by backward-propagation of the value function (17). To ensure convergence, we

employ the line search strategy as in [2]. We compute the control law as $\delta\hat{\mathbf{u}}_k = \alpha\mathbf{I}_k + \mathbf{L}_k\delta\mathbf{z}_k^x$. Initially $\alpha = 1$, then decrease it until the expected cost is smaller than the previous one. In *Forward propagation* (Step 6), we apply the control sequence from last step and obtain a new nominal trajectory for the next iteration. In *Convergence condition* (Step 7), we set a threshold on the accumulated cost $J^*$ such that when $J^\pi < J^*$, the algorithm is terminated with the optimized state and control trajectory. In *Interaction condition* (Step 8), when the state covariance $\boldsymbol{\Sigma}_k$ exceeds a threshold $\boldsymbol{\Sigma}_{tol}$, we sample new trajectories from the physical system using the control obtained in step 5, and go back to step 2 to learn a more accurate model. The old GP training data points are removed from the training set to keep its size fixed. Finally in *Nominal trajectory update* (step 9), the trajectory obtained in Step 6 or 8 becomes the new nominal trajectory for the next iteration. An simple illustration of the algorithm is shown in Fig. 3a. Intuitively, PDDP requires interactions with the physical systems only if the GP model no longer represents the true dynamics around the nominal trajectory.

---

**Given**: A system with unknown dynamics, target states
**Goal** : An optimized trajectory of state and control

**1** Generate $N$ state trajectories by applying random control sequences to the physical system (1);
**2** Obtain state and control training pairs from sampled trajectories and optimize the hyper-parameters of GP (8);
**3** **for** $i = 1$ **to** $I_{max}$ **do**
**4**     Compute a linear approximation of the dynamics along $(\bar{\mathbf{z}}_k^x, \bar{\mathbf{u}}_k)$ (10);
**5**     Backpropagate in time to get the locally optimal control $\hat{\mathbf{u}}_k = \bar{\mathbf{u}}_k + \delta\hat{\mathbf{u}}_k$ and value function $V(\mathbf{z}_k^x, k)$ according to (16) (17);
**6**     Forward propagate the dynamics (9) by applying the optimal control $\hat{\mathbf{u}}_k$, obtain a new trajectory $(\mathbf{z}_k^x, \mathbf{u}_k)$;
**7**     **if** *Converge* **then** Break the **for** loop;
**8**     **if** $\boldsymbol{\Sigma}_k > \boldsymbol{\Sigma}_{tol}$ **then** apply the optimal control to the original physical system to generate a new nominal trajectory $(\mathbf{z}_k^x, \mathbf{u}_k)$ and $N - 1$ additional trajectories by applying small variations of the learned controller, update the GP training set and go back to step 2;
**9**     Set $\bar{\mathbf{z}}_k^x = \mathbf{z}_k^x$, $\bar{\mathbf{u}}_k = \mathbf{u}_k$ and $i = i + 1$, go back to step 4;
**10 end**
**11** Apply the optimized controller to the physical system, obtain the optimized trajectory.

**Algorithm 1:** PDDP algorithm

---

## 2.7 Computational complexity

*Dynamics propagation*: The major computational effort is devoted to GP inferences. In particular, the complexity of one-step moment matching (2.2) is $\mathcal{O}\big((N)^2 n^2(n+m)\big)$ [14], which is fixed during the iterative process of PDDP. We found a small number of sampled trajectories ($N \leq 5$) are able to provide good performances for a system of moderate size (6-12 state dimensions). However, for higher dimensional problems, sparse or local approximation of GP (e.g. [11][18][19], etc) may be used to reduce the computational cost of GP dynamics propagation.

*Controller learning*: According to (16), learning policy parameters $\mathbf{I}_k$ and $\mathbf{L}_k$ requires computing the inverse of $\mathbf{Q}_k^{uu}$, which has the computational complexity of $\mathcal{O}(m^3)$, where $m$ is the dimension of control input. As a local trajectory optimization method, PDDP offers comparable scalability to the classical DDP.

## 2.8 Relation to existing works

Here we summarize the novel features of PDDP in comparison with some notable DDP-related frameworks for stochastic systems (see also Table 1). First, PDDP shares some similarities with the belief space iLQG [6] framework, which approximates the belief dynamics using an extended Kalman filter. Belief space iLQG assumes a dynamics model is given and the stochasticity comes from the process noises. PDDP, however, is a data-driven approach that learns the dynamics models and controls from sampled data, and it takes into account model uncertainties by using GPs. Second, PDDP is also comparable with iLQG-LD [5], which applies Locally Weighted Projection Regression (LWPR) to represent the dynamics. iLQG-LD does not incorporate model uncertainty therefore requires a large amount of data to learn an accurate model. Third, PDDP does not suffer from the

high computational cost of finite differences used to numerically compute the first-order expansions [2][6] and second-order expansions [4] of the underlying stochastic dynamics. PDDP computes Jacobian matrices analytically (11).

| | PDDP | Belief space iLQG[6] | iLQG-LD[5] | iLQG[2]/sDDP[4] |
|---|---|---|---|---|
| State | $\boldsymbol{\mu}_k, \boldsymbol{\Sigma}_k$ | $\boldsymbol{\mu}_k, \boldsymbol{\Sigma}_k$ | $\mathbf{x}_k$ | $\mathbf{x}_k$ |
| Dynamics model | Unknown | Known | Unknown | Known |
| Linearization | Analytic Jacobian | Finite differences | Analytic Jacobian | Finite differences |

Table 1: Comparison with DDP-related frameworks

## 3 Experimental Evaluation

We evaluate the PDDP framework using two nontrivial simulated examples: i) cart-double inverted pendulum swing-up; ii) six-link robotic arm reaching. We also compare the learning efficiency of PDDP with the classical DDP [1] and PILCO [13][14]. All experiments were performed in MATLAB.

### 3.1 Cart-double inverted pendulum swing-up

Cart-Double Inverted Pendulum (CDIP) swing-up is a challenging control problem because the system is highly underactuated with 3 degrees of freedom and only 1 control input. The system has 6 state-dimensions (cart position/velocity, link 1,2 angles and angular velocities). The swing-up problem is to find a sequence of control input to force both pendulums from initial position $(\pi,\pi)$ to the inverted position $(2\pi,2\pi)$. The balancing task requires the velocity of the cart, angular velocities of both pendulums to be zero. We sample 4 initial trajectories with time horizon $H = 50$. The CDIP swing-up problem has been solved by two controllers for swing-up and balancing, respectively [20]. PILCO [14] is one of the few RL methods that is able to complete this task without knowing the dynamics. The results are shown in Fig.1.

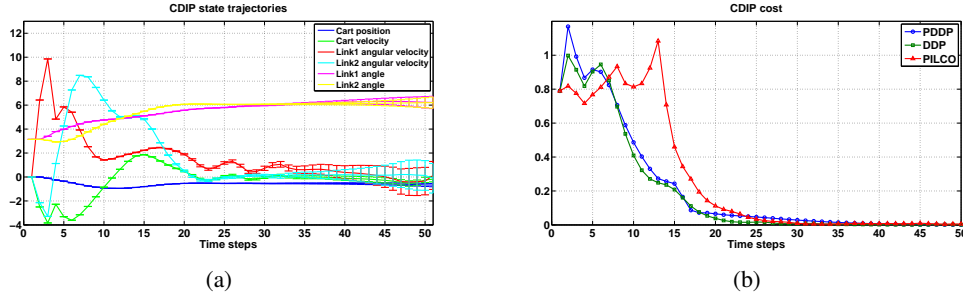

Figure 1: Results for the CDIP task. (a) Optimized state trajectories of PDDP. Solid lines indicate means, errorbars indicate variances. (b) Cost comparison of PDDP, DDP and PILCO. Costs (eq. 13) were computed based on sampled trajectories by applying the final controllers.

### 3.2 Six-link robotic arm

The six-link robotic arm model consist of six links of equal length and mass, connected in an open chain with revolute joints. The system has 6 degrees of freedom, and 12 state dimensions (angle and angular velocity for each joint). The goal for the first 3 joints is to move to the target angle $\frac{\pi}{4}$ and for the rest 3 joints to $-\frac{\pi}{4}$. The desired velocities for all 6 joints are zeros. We sample 2 initial trajectories with time horizon $H = 50$. The results are shown in Fig. 2.

### 3.3 Comparative analysis

**DDP**: Originally introduced in the 70's, the classical DDP [1] is still one of the most effective and efficient trajectory optimization approaches. The major differences between DDP and PDDP can

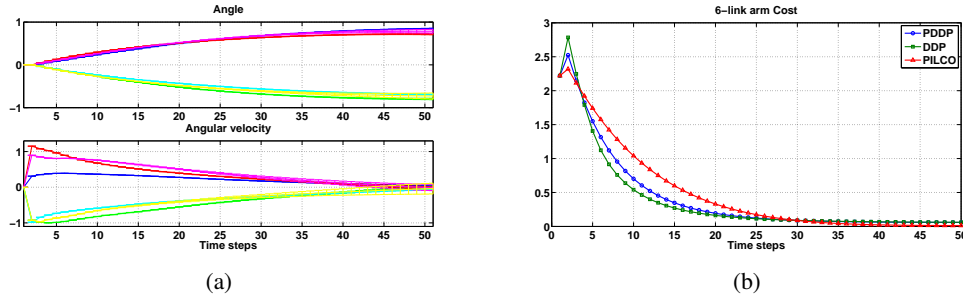

Figure 2: Results for the 6-link arm task. (a) Optimized state trajectories of PDDP. Solid lines indicate means, errorbars indicate variances. (b) Cost comparison of PDDP, DDP and PILCO. Costs (eq. 13) were computed based on sampled trajectories by applying the final controllers.

be summarized as follow: firstly, DDP relies on a given accurate dynamics model, while PDDP is a data-driven framework that learns a locally accurate model by forward sampling; secondly, DDP does not deal with model uncertainty, PDDP takes into account model uncertainty using GPs and perform local dynamic programming in Gaussian belief spaces; thirdly, generally in applications of DDP linearizations are performed using finite differences while in PDDP Jacobian matrices are computed analytically (11).

**PILCO**: The recently proposed PILCO [14] framework has demonstrated state-of-the-art learning efficiency compared with other methods such as [21][22]. The proposed PDDP is different from PILCO in several ways. Firstly, based on local linear approximation of dynamics and quadratic approximation of the value function, PDDP finds linear, time-varying feedforward and feedback policy, PILCO requires an a priori policy parameterization and an extra optimization solver. Secondly, PDDP keeps a fixed size of training data for GP inferences, while PILCO adds new data to the training set after each trial (recently, the authors applied sparse GP approximation [19] in an improved version of PILCO when the data size reached a threshold). Thirdly, by using the Gaussian belief augmented state and cost function (13), PDDP's exploration scheme is balanced between the distance from the target and the variance of the state. PILCO employs a saturating cost function which leads to automatic explorations in the high-variance regions in the early stages of learning.

In both tasks, PDDP, DDP and PILCO bring the system to the desired states. The resulting trajectories for PDDP are shown in Fig.1a and 2a. The reason for low variances of some optimized trajectories is that during final stage of learning, interactions with the physical systems (forward samplings using the locally optimal controller) would reduce the variances significantly. The costs are shown in Fig. 1b and 2b. For both tasks, PDDP and DDP performs similarly and slightly different from PILCO in terms of cost reduction. The major reasons for this difference are: i) different cost functions used by these methods; ii) we did not impose any convergence condition for the optimized trajectories on PILCO. We now compare PDDP with DDP and PILCO in terms of data-efficiency and controller learning speed.

*Data-efficiency*: As shown in Fig.4a, in both tasks, PDDP performs slightly worse than PILCO in terms of data-efficiency based on the number of interactions required with the physical systems. For the systems used for testing, PDDP requires around $15\% - 25\%$ more interactions than PILCO. The number of interactions indicates the amount of sampled trajectories required from the physical system. At each trial we sample $N$ trajectories from the physical systems (algorithm 1). Possible reasons for the slightly worse performances are: i) PDDP's policy is linear which is restrictive, while PILCO yields nonlinear policy parameterizations; ii) PDDP's exploration scheme is more conservative than PILCO in the early stages of learning. We believe PILCO is the most data-efficient framework for these tasks. However, PDDP is able to offer close performances thanks to the probabilistic representation of the dynamics as well as the use of Gaussian belief augmented state.

*Learning speed*: In terms of total computational time required to obtain the final controller, PDDP outperforms PILCO significantly as shown in Fig.4b. For the 6 and 12 dimensional systems used for testing, PILCO requires an iterative method (e.g.,CG or BFGS) to solve high dimensional optimization problems (depending on the policy parameterization), while PDDP computes local optimal controls (16) without an extra optimizer. In terms of computational time per iteration, as shown in

Fig.3b, PDDP is slower than the classical DDP due to the high computational cost of GP dynamics propagations. However, for DDP, the time dedicated to linearizing the dynamics model is around $70\% - 90\%$ of the total time per iteration for the two tasks considered in this work. PDDP avoids the high computational cost of finite differences by evaluating all Jacobian matrices analytically, the time dedicated to linearization is less than $10\%$ of the total time per iteration.

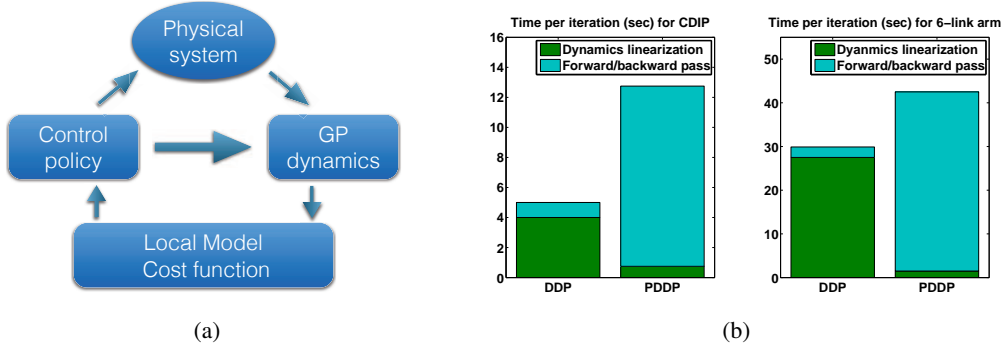

|            |            |
| :--------: | :--------: |
|    (a)     |    (b)     |

Figure 3: (a) An intuitive illustration of the PDDP framework. (b) Comparison of PDDP and DDP in terms of the computational time per iteration (in seconds) for the CDIP (left subfigure) and 6-link arm (right subfigure) tasks. Green indicates time for performing linearization, cyan indicates time for forward and backward sweeps (Sec. 2.6).

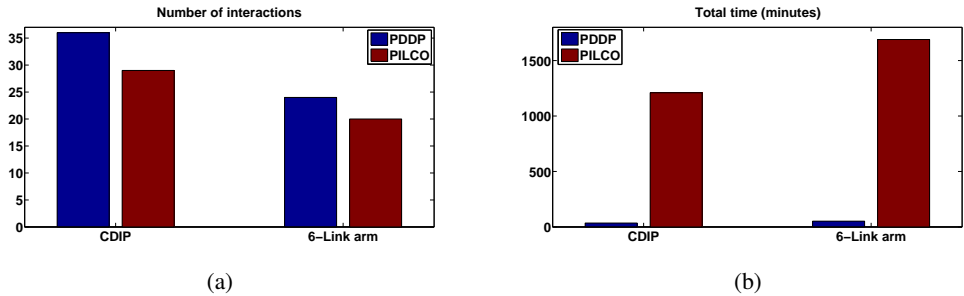

|            |            |
| :--------: | :--------: |
|    (a)     |    (b)     |

Figure 4: Comparison of PDDP and PILCO in terms of data-efficiency and controller learning speed. (a) Number of interactions with the physical systems required to obtain the final results in Fig. 1 and 2. (b) Total computational time (in minutes) consumed to obtain the final controllers.

## 4  Conclusions

In this work we have introduced a probabilistic model-based control and trajectory optimization method for systems with unknown dynamics based on Differential Dynamic Programming (DDP) and Gaussian processes (GPs), called Probabilistic Differential Dynamic Programming (PDDP). PDDP takes model uncertainty into account explicitly by representing the dynamics using GPs and performing local Dynamic Programming in Gaussian belief spaces. Based on the quadratic approximation of the value function, PDDP yields a linear, locally optimal control policy and features a more efficient control improvement scheme compared with typical gradient-based policy search methods. Thanks to the probabilistic representation of the dynamics, PDDP offers reasonable data-efficiency comparable to a state of the art GP-based policy search method [14]. In general, local trajectory optimization is a powerful approach to challenging control and RL problems. Due to its model-based nature, model inaccuracy has always been the major obstacle for advanced applications. Grounded on the solid developments of classical trajectory optimization and Bayesian machine learning, the proposed PDDP has demonstrated encouraging performance and potential for many applications.

**Acknowledgments**

This work was partially supported by a National Science Foundation grant NRI-1426945.

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
