[Supplementary Material]

# Supplementary Material: Probabilistic Differential Dynamic Programming

**Yunpeng Pan** and **Evangelos A. Theodorou**
Daniel Guggenheim School of Aerospace Engineering
Institute for Robotics and Intelligent Machines
Georgia Institute of Technology
Atlanta, GA 30332
ypan37@gatech.edu, evangelos.theodorou@ae.gatech.edu

## Abstract

This is the supplementary document for the paper on Probabilistic Differential Dynamic Programming (PDDP). It includes derivations for the probabilistic representation of the stochastic dynamics, the linearization of the dynamics model and the cost function formulation.

## 1 Problem formulation

We consider a general unknown stochastic system described by the following differential equation

$$d\mathbf{x} = \mathbf{f}(\mathbf{x}, \mathbf{u})dt + \mathbf{C}(\mathbf{x}, \mathbf{u})d\omega, \qquad \mathbf{x}(t_0) = \mathbf{x}_0, \qquad d\omega \sim \mathcal{N}(0, \mathbf{\Sigma}_\omega), \tag{1}$$

where $\mathbf{x} \in \mathbb{R}^n$ is the state, $\mathbf{u} \in \mathbb{R}^m$ is the control, $t$ is time and $\omega \in \mathbb{R}^p$ is standard Brownian motion noise. The trajectory optimization problem is defined as finding a sequence of state and controls that minimize the expected cost

$$J^\pi(\mathbf{x}(t_0)) = \mathbb{E}\left[ h\Big(\mathbf{x}(T)\Big) + \int_{t_0}^T \mathcal{L}\Big(\mathbf{x}(t), \pi(\mathbf{x}(t)), t\Big)dt \right], \tag{2}$$

where $h(\mathbf{x}(T))$ is the terminal cost, $\mathcal{L}(\mathbf{x}(t), \pi(\mathbf{x}(t)), t)$ is the instantaneous cost rate, $\mathbf{u}(t) = \pi(\mathbf{x}(t))$ is the control policy. The cost $J^\pi(\mathbf{x}(t_0))$ is defined as the expectation of the total cost accumulated from $t_0$ to $T$. For the rest of our analysis, we denote $\mathbf{x}_k = \mathbf{x}(k)$ in discrete-time where $k = 0, 1, ..., H$ is the time step, we use this subscript rule for other variables as well.

## 2 Probabilistic model learning

The continuous functional mapping from state-control pair $\tilde{\mathbf{x}} = (\mathbf{x}, \mathbf{u}) \in \mathbb{R}^{n+m}$ to state transition $d\mathbf{x}$ can be viewed as an inference with the goal of inferring $d\mathbf{x}$ given $\tilde{\mathbf{x}}$. We view this inference as a nonlinear regression problem. In this subsection, we introduce the Gaussian processes (GP) approach to learning the dynamics model in (1). A GP is defined as a collection of random variables, any finite number subset of which have a joint Gaussian distribution. Given a sequence of state-control pair $\tilde{\mathbf{X}} = \{(\mathbf{x}_0, \mathbf{u}_0), \ldots (\mathbf{x}_H, \mathbf{u}_H)\}$, and the corresponding state transition $d\mathbf{X} = \{d\mathbf{x}_0, \ldots, d\mathbf{x}_H\}$, a GP is completely defined by a mean function and a covariance function. The joint distribution of the observed output and the output corresponding to a given test state-control pair $\tilde{\mathbf{X}}^* = (\mathbf{x}^*, \mathbf{u}^*)$ can be written as

$$p\left( \begin{array}{c} d\mathbf{X} \\ d\mathbf{x}^* \end{array} \right) \sim \mathcal{N}\left(0, \left[ \begin{array}{cc} \mathbf{K}(\tilde{\mathbf{X}}, \tilde{\mathbf{X}}) + \sigma_n \mathbf{I} & \mathbf{K}(\tilde{\mathbf{X}}, \tilde{\mathbf{x}}^*) \\ \mathbf{K}(\tilde{\mathbf{x}}^*, \tilde{\mathbf{X}}) & \mathbf{K}(\tilde{\mathbf{x}}^*, \tilde{\mathbf{x}}^*) \end{array} \right] \right). \tag{3}$$

The covariance of this multivariate Gaussian distribution is defined via a kernel matrix $\mathbf{K}(\mathbf{x}_i, \mathbf{x}_j)$. In particular, in this paper we consider the Gaussian kernel

$$\mathbf{K}(\mathbf{x}_i, \mathbf{x}_j) = \sigma_s^2 \exp(-\frac{1}{2}(\mathbf{x}_i - \mathbf{x}_j)^\mathrm{T} \mathbf{W}(\mathbf{x}_i - \mathbf{x}_j)) + \sigma_n^2, \tag{4}$$

with $\sigma_s, \sigma_n, \mathbf{W}$ the hyper-parameters of the GP. The kernel function can be interpreted as a similarity measure of random variables. More specifically, if the training pairs $\tilde{\mathbf{X}}_i$ and $\tilde{\mathbf{X}}_j$ are close to each other in the kernel space, their output $\mathrm{d}\mathbf{x}_i$ and $\mathrm{d}\mathbf{x}_j$ are highly correlated.

The posterior distribution, which is also a Gaussian distribution, can be obtained by constraining the joint distribution to contain the output $\mathrm{d}\mathbf{x}^*$ that are consistent with the observations. Assuming independent outputs (no correlation between each output dimension) and given test input $\tilde{\mathbf{x}}_k = [\mathbf{x}_k, \mathbf{u}_k]$ at time $k$. The one-step prediction of dynamics based on GP can be evaluated as

$$\mathrm{p}(\mathrm{d}\mathbf{x}_k | \tilde{\mathbf{x}}_k) \sim \mathcal{N}(\mathrm{d}\boldsymbol{\mu}_k, \mathrm{d}\boldsymbol{\Sigma}_k), \tag{5}$$

where the mean and variance are given by

$$\begin{aligned}
\mathrm{d}\boldsymbol{\mu}_k =& \mathbb{E}[\mathrm{d}\mathbf{x}_k] = \mathbf{K}(\tilde{\mathbf{x}}_k, \tilde{\mathbf{X}})(\mathbf{K}(\tilde{\mathbf{X}}, \tilde{\mathbf{X}}) + \sigma_n \mathbf{I})^{-1} \mathrm{d}\mathbf{X}, \\
\mathrm{d}\boldsymbol{\Sigma}_k =& \mathrm{Var}[\mathrm{d}\mathbf{x}_k] = \mathbf{K}(\tilde{\mathbf{x}}_k, \tilde{\mathbf{x}}_k) - \mathbf{K}(\tilde{\mathbf{x}}_k, \tilde{\mathbf{X}})(\mathbf{K}(\tilde{\mathbf{X}}, \tilde{\mathbf{X}}) + \sigma_n \mathbf{I})^{-1} \mathbf{K}(\tilde{\mathbf{X}}, \tilde{\mathbf{x}}_k)
\end{aligned} \tag{6}$$

where $\mathrm{d}\boldsymbol{\mu}_k$ and $\mathrm{d}\boldsymbol{\Sigma}_k$ are predictive mean and variance of the state transition, respectively. Therefore, the state distribution at $k+1$ would be:

$$\mathrm{p}(\mathbf{x}_k) \sim \mathcal{N}(\boldsymbol{\mu}_k, \boldsymbol{\Sigma}_k), \tag{7}$$

where the state mean an variance are

$$\boldsymbol{\mu}_{k+1} = \mathbf{x}_k + \mathrm{d}\boldsymbol{\mu}_k, \qquad \boldsymbol{\Sigma}_k = \mathrm{d}\boldsymbol{\Sigma}_k. \tag{8}$$

When propagating the GP-based dynamics over a trajectory of time horizon $H$, the input state $\mathbf{x}_k$ becomes uncertain with Gaussian distribution, where $k = 1, ..., H$ (the initial state $\mathbf{x}_0$ is deterministic). Thus the distribution over state transition can be computed as:

$$\mathrm{p}(\mathrm{d}\mathbf{x}_k) = \int \int \mathrm{p}(\mathbf{f}(\mathbf{x}_k, \mathbf{u}_k) | \mathbf{x}_k, \mathbf{u}_k) \mathrm{p}(\mathbf{x}_k, \mathbf{u}_k) \mathrm{d}\mathbf{x}_k \mathrm{d}\mathbf{u}_k. \tag{9}$$

Generally, the above distribution cannot be computed analytically because the nonlinear mapping of input Gaussian distributions lead to non-Gaussian predictive distributions. However, the predictive distribution can be approximated by a Gaussian. Thus the state distribution at $t+1$ is also a Gaussian

$$\boldsymbol{\mu}_{k+1} = \boldsymbol{\mu}_k + \mathrm{d}\boldsymbol{\mu}_k, \qquad \boldsymbol{\Sigma}_{k+1} = \boldsymbol{\Sigma}_k + \mathrm{d}\boldsymbol{\Sigma}_k + \mathrm{Cov}[\mathbf{x}_k, \mathrm{d}\mathbf{x}_k] + \mathrm{Cov}[\mathrm{d}\mathbf{x}_k, \mathbf{x}_k]. \tag{10}$$

In order to obtain the distribution over state $\mathcal{N}(\boldsymbol{\mu}_{k+1}, \boldsymbol{\Sigma}_{k+1})$. Firstly, we compute the joint distribution over state-control pair $\mathrm{p}(\tilde{\mathbf{x}}_k) = \mathrm{p}(\mathbf{x}_k, \mathbf{u}_k)$ as follow

$$\mathrm{p}\left(\begin{array}{c} \mathbf{x}_k \\ \mathbf{u}_k \end{array}\right) \sim \mathcal{N}\left(\begin{array}{c} \boldsymbol{\mu}_k \\ \mathbb{E}[\mathbf{u}_k] \end{array}, \left[\begin{array}{cc} \boldsymbol{\Sigma}_k & \mathrm{Cov}[\mathbf{x}_k, \mathbf{u}_k] \\ \mathrm{Cov}[\mathbf{u}_k, \mathbf{x}_k] & \mathrm{Cov}[\mathbf{u}_k] \end{array}\right]\right) \tag{11}$$

where $\mathbb{E}[\mathbf{u}_k]$ and $\mathrm{Cov}[\mathbf{u}_k]$ are mean and covariance of the distribution over control policy $\mathrm{p}(\mathbf{u}_k)$. To simplify notation, we denote the mean and covariance of the above distribution as $\mathrm{p}(\tilde{\mathbf{x}}_k) \sim \mathcal{N}(\tilde{\boldsymbol{\mu}}_k, \tilde{\boldsymbol{\Sigma}}_k)$. Since the control policy is a linear function of the Gaussian belief augmented state in this paper, the control is actually deterministic.

Given the input joint distribution $\mathrm{p}(\tilde{\mathbf{x}}_k)$, we will compute the predictive distribution of state transition $\mathrm{p}(\mathrm{d}\mathbf{x}_k)$. The predictive mean can be computed using the law of iterated expectations (Fubini's

theorem)

$$
\begin{aligned}
\mathrm{d}\boldsymbol{\mu}_k &= \int\int\int \mathbf{f}(\mathbf{x}_k, \mathbf{u}_k)\mathrm{p}(\mathbf{x}_k, \mathbf{u}_k)\mathrm{d}\mathbf{f}\mathrm{d}\mathbf{x}_k\mathrm{d}\mathbf{u}_k \\
&= \int\int \mathbf{f}(\tilde{\mathbf{x}}_k)\mathrm{p}(\tilde{\mathbf{x}}_k)\mathrm{d}\mathbf{f}\mathrm{d}\tilde{\mathbf{x}}_k\mathrm{d}t \\
&= \mathbb{E}_{\mathbf{f},\tilde{\mathbf{x}}_k}[\mathbf{f}|\tilde{\boldsymbol{\mu}}_k, \tilde{\boldsymbol{\Sigma}}_k] \\
&= \mathbb{E}_{\tilde{\mathbf{x}}_k}\Big[\mathbb{E}_{\mathbf{f}}[\mathbf{f}(\tilde{\mathbf{x}}_k)|\tilde{\mathbf{x}}_k]|\tilde{\boldsymbol{\mu}}_k, \tilde{\boldsymbol{\Sigma}}_k\Big]\mathrm{d}t \\
&= \int\Big(\mathbf{K}(\tilde{\mathbf{x}}_k, \tilde{\mathbf{X}})(\mathbf{K}(\tilde{\mathbf{X}}, \tilde{\mathbf{X}}) + \sigma_n^2\mathbf{I})^{-1}\mathrm{d}\mathbf{X}\Big)\mathcal{N}(\tilde{\mathbf{x}}_k|\tilde{\boldsymbol{\mu}}_k, \tilde{\boldsymbol{\Sigma}}_k)\mathrm{d}\tilde{\mathbf{x}}_k \\
&= \underbrace{\Big(\mathbf{K}(\tilde{\mathbf{X}}, \tilde{\mathbf{X}}) + \sigma_n^2\mathbf{I})^{-1}\mathrm{d}\mathbf{X}\Big)^{\mathrm{T}}}_{\Psi}\underbrace{\int\mathbf{K}(\tilde{\mathbf{X}}, \tilde{\mathbf{x}}_k)\mathcal{N}(\tilde{\mathbf{x}}_k|\tilde{\boldsymbol{\mu}}_k, \tilde{\boldsymbol{\Sigma}}_k)\mathrm{d}\tilde{\mathbf{x}}_k}_{\mathbf{q}} \\
&= \Psi^{\mathrm{T}}\mathbf{q}_k \qquad\qquad\qquad\qquad\qquad\qquad\qquad\qquad\qquad\qquad (12)
\end{aligned}
$$

where $\Psi \in \mathbb{R}^{N\times n}$ and $\mathbf{q}_k = [q_{k1}, \ldots, q_{kn}]^{\mathrm{T}} \in \mathbb{R}^N$ with each element

$$
\begin{aligned}
q_{ki} &= \int\mathbf{K}(\tilde{\mathbf{X}}_i, \tilde{\mathbf{x}}_k)\mathcal{N}(\tilde{\mathbf{x}}_k|\tilde{\boldsymbol{\mu}}_k, \tilde{\boldsymbol{\Sigma}}_k)\mathrm{d}\tilde{\mathbf{x}}_k \\
&= \alpha^2|\tilde{\boldsymbol{\Sigma}}_k + \mathbf{W}|^{\frac{1}{2}}\exp\Big(-\frac{1}{2}(\tilde{\mathbf{X}}_i - \tilde{\boldsymbol{\mu}}_k)^{\mathrm{T}}(\tilde{\boldsymbol{\Sigma}}_k + \Lambda)^{-1}(\tilde{\mathbf{X}}_i - \tilde{\boldsymbol{\mu}}_k)\Big). \quad (13)
\end{aligned}
$$

Next, we compute the predictive covariance matrix

$$
\mathrm{Cov}(\mathrm{d}\mathbf{x}_k|\tilde{\mathbf{x}}_k) = \begin{bmatrix} \mathrm{Var}(\mathrm{d}\mathbf{x}_{k1}) & \ldots & \mathrm{Cov}(\mathrm{d}\mathbf{x}_{kn}, \mathrm{d}\mathbf{x}_{k1}) \\ \vdots & \ddots & \vdots \\ \mathrm{Cov}(\mathrm{d}\mathbf{x}_{k1}, \mathrm{d}\mathbf{x}_{kn}) & \ldots & \mathrm{Var}(\mathrm{d}\mathbf{x}_{kn}) \end{bmatrix} \qquad (14)
$$

where the variance terms can be obtained as

$$
\begin{aligned}
\mathrm{Var}(\mathrm{d}\mathbf{x}_k) =& \mathbb{E}_{\tilde{\mathbf{x}}_k}\big[\mathrm{Var}(\mathbf{f}(\tilde{\mathbf{x}}_k)|\tilde{\boldsymbol{\mu}}_k, \tilde{\boldsymbol{\Sigma}}_k)\big] + \mathrm{Var}\Big(\mathbb{E}_{\mathbf{f}}\big[\mathbf{f}(\tilde{\mathbf{x}}_k)|\tilde{\boldsymbol{\mu}}_k, \tilde{\boldsymbol{\Sigma}}_k\big]\Big) \\
=& \mathbb{E}_{\tilde{\mathbf{x}}_k}\big[\mathrm{Var}(\mathrm{d}\mathbf{x}_k)\big] + \Big(\mathbb{E}_{\tilde{\mathbf{x}}_k}\big[(\mathrm{d}\mathbf{x}_k)^2\big] - \mathbb{E}_{\tilde{\mathbf{x}}_k}\big[\mathrm{d}\mathbf{x}_k\big]^2\Big) \\
=& \int\Big(\mathbf{K}(\tilde{\mathbf{x}}_k, \tilde{\mathbf{x}}_k) - \mathbf{K}(\tilde{\mathbf{x}}_k, \tilde{\mathbf{X}})(\mathbf{K}(\tilde{\mathbf{X}}, \tilde{\mathbf{X}}) + \sigma_n^2\mathbf{I})^{-1}\mathbf{K}(\tilde{\mathbf{X}}, \tilde{\mathbf{x}}_k)\Big)\mathrm{p}(\tilde{\mathbf{x}}_k)\mathrm{d}\tilde{\mathbf{x}}_k \\
& + \int\Big(\mathbf{K}(\tilde{\mathbf{x}}_k, \tilde{\mathbf{X}})(\mathbf{K}(\tilde{\mathbf{X}}, \tilde{\mathbf{X}}) + \sigma_n^2\mathbf{I})^{-1}\mathrm{d}\mathbf{X}\Big)^2\mathrm{p}(\tilde{\mathbf{x}}_k)\mathrm{d}\tilde{\mathbf{x}}_k \\
& - \Big(\big(\mathbf{K}(\tilde{\mathbf{X}}, \tilde{\mathbf{X}}) + \sigma_n^2\mathbf{I})^{-1}\mathrm{d}\mathbf{X}\big)^{\mathrm{T}}\int\mathbf{K}(\tilde{\mathbf{x}}_k, \tilde{\mathbf{X}})\mathcal{N}(\tilde{\mathbf{x}}_k|\tilde{\boldsymbol{\mu}}_k, \tilde{\boldsymbol{\Sigma}}_k)\mathrm{d}\tilde{\mathbf{x}}_k\Big)^2. \quad (15)
\end{aligned}
$$

The last term in the above equation can be represented by $\Psi$ and $\mathbf{q}$ defined earlier, then the equation becomes

$$
\begin{aligned}
\mathrm{Var}(\mathrm{d}\mathbf{x}_k) =& \int\Big(\mathbf{K}(\tilde{\mathbf{x}}_k, \tilde{\mathbf{x}}_k) - \mathbf{K}(\tilde{\mathbf{x}}_k, \tilde{\mathbf{X}})(\mathbf{K}(\tilde{\mathbf{X}}, \tilde{\mathbf{X}}) + \sigma_n^2\mathbf{I})^{-1}\mathbf{K}(\tilde{\mathbf{X}}, \tilde{\mathbf{x}}_k)\Big)\mathrm{p}(\tilde{\mathbf{x}}_k)\mathrm{d}\tilde{\mathbf{x}}_k \\
& + \int\mathbf{K}(\tilde{\mathbf{x}}_k, \tilde{\mathbf{X}})\Psi\Psi^{\mathrm{T}}\mathbf{K}(\tilde{\mathbf{X}}, \tilde{\mathbf{x}}_k)\mathrm{p}(\tilde{\mathbf{x}}_k)\mathrm{d}\tilde{\mathbf{x}}_k - \big(\Psi^{\mathrm{T}}\mathbf{q}_k\big)^2. \quad (16)
\end{aligned}
$$

Re-arrange the above expressions by pulling the terms that are independent of $\tilde{\mathbf{x}}_k$ out of the integrals:

$$
\begin{aligned}
\mathrm{Var}(\mathrm{d}\mathbf{x}_k) =& \sigma_s^2 - \mathrm{tr}\Big((\mathbf{K}(\tilde{\mathbf{X}}, \tilde{\mathbf{X}}) + \sigma_n^2\mathbf{I})^{-1}\int\Big(\mathbf{K}(\tilde{\mathbf{X}}, \tilde{\mathbf{x}}_k)\mathbf{K}(\tilde{\mathbf{x}}_k, \tilde{\mathbf{X}})\Big)\mathrm{p}(\tilde{\mathbf{x}}_k)\mathrm{d}\tilde{\mathbf{x}}_k\Big) \\
& + \Psi^{\mathrm{T}}\underbrace{\Big(\int\mathbf{K}(\tilde{\mathbf{X}}, \tilde{\mathbf{x}}_k)\mathbf{K}(\tilde{\mathbf{x}}_k, \tilde{\mathbf{X}})\mathrm{p}(\tilde{\mathbf{x}}_k)\mathrm{d}\tilde{\mathbf{x}}_k\Big)}_{\Phi_k}\Psi - \big(\Psi^{\mathrm{T}}\mathbf{q}_k\big)^2 \\
=& \sigma_s^2 - \mathrm{tr}\Big((\mathbf{K}(\tilde{\mathbf{X}}, \tilde{\mathbf{X}}) + \sigma_n^2\mathbf{I})^{-1}\Phi_k\Big) + \Psi^{\mathrm{T}}\Phi_k\Psi - \big(\Psi^{\mathrm{T}}\mathbf{q}_k\big)^2, \quad (17)
\end{aligned}
$$

where the integral terms $\Phi_k$ can be evaluated as

$$
\begin{aligned}
\Phi_{ij} &= \int \mathbf{K}(\tilde{\mathbf{X}}, \tilde{\mathbf{x}}_k) \mathbf{K}(\tilde{\mathbf{x}}_k, \tilde{\mathbf{X}}) \mathrm{p}(\tilde{\mathbf{x}}_k) \mathrm{d}\tilde{\mathbf{x}}_k \\
&= \frac{\mathbf{K}(\tilde{\mathbf{X}}_i, \tilde{\boldsymbol{\mu}}_k) \mathbf{K}(\tilde{\mathbf{X}}_j, \tilde{\boldsymbol{\mu}}_k)}{|2\tilde{\boldsymbol{\Sigma}}_k(\mathbf{W}_i^{-1} + \mathbf{W}_j^{-1}) + I|^{\frac{1}{2}}} \exp\Big(\big(\frac{1}{2}(\frac{\mathbf{W}_j}{\mathbf{W}_i + \mathbf{W}_j}\tilde{\mathbf{x}}_i + \frac{\mathbf{W}_i}{\mathbf{W}_i + \mathbf{W}_j}\tilde{\mathbf{x}}_j) - \tilde{\boldsymbol{\mu}}_k\big)^{\mathrm{T}} \\
&\quad \big(\tilde{\boldsymbol{\Sigma}} + \frac{1}{2}\mathbf{W}\big)^{-1}\tilde{\boldsymbol{\Sigma}}_k \mathbf{W}^{-1}\big(\frac{1}{2}(\frac{\mathbf{W}_j}{\mathbf{W}_i + \mathbf{W}_j}\tilde{\mathbf{x}}_i + \frac{\mathbf{W}_i}{\mathbf{W}_i + \mathbf{W}_j}\tilde{\mathbf{x}}_j) - \tilde{\boldsymbol{\mu}}_k\big)\Big),
\end{aligned}
\tag{18}
$$

where $\mathbf{W}_i, \mathbf{W}_j$ are the kernel parameters corresponding to output dimension $i$ and $j$, respectively. The cross covariance terms can be obtained by

$$
\mathrm{Cov}(\mathrm{d}\mathbf{x}_{ki}, \mathrm{d}\mathbf{x}_{kj}) = \mathbb{E}_{\tilde{\mathbf{x}}_k}\big[\mathrm{d}\mathbf{x}_{ki}\mathrm{d}\mathbf{x}_{kj}\big] - \mathbb{E}_{\tilde{\mathbf{x}}_k}\big[\mathrm{d}\mathbf{x}_{ki}\big]\mathbb{E}_{\tilde{\mathbf{x}}_k}\big[\mathrm{d}\mathbf{x}_{kj}\big]
\tag{19}
$$

Similarly, it can be found that the first term is

$$
\mathbb{E}_{\tilde{\mathbf{x}}_k}\big[\mathrm{d}\mathbf{x}_{ki}\mathrm{d}\mathbf{x}_{kj}\big] = \Psi_i^{\mathrm{T}}\Phi_k\Psi_j,
\tag{20}
$$

where

$$
\Psi_i = \mathbf{K}(\tilde{\mathbf{X}}, \tilde{\mathbf{X}}) + \sigma_n^2\mathbf{I})^{-1}\mathrm{d}\mathbf{X}_i, \quad \Psi_j = \mathbf{K}(\tilde{\mathbf{X}}, \tilde{\mathbf{X}}) + \sigma_n^2\mathbf{I})^{-1}\mathrm{d}\mathbf{X}_j.
\tag{21}
$$

Therefore

$$
\mathrm{Cov}[\mathrm{d}\mathbf{x}_{ki}, \mathrm{d}\mathbf{x}_{kj}] = \Psi_i^{\mathrm{T}}\Phi_k\Psi_j - (\Psi_i^{\mathrm{T}}\mathbf{q}_k)^{\mathrm{T}}(\Psi_j^{\mathrm{T}}\mathbf{q}_k).
\tag{22}
$$

The input-output cross-covariances can be obtained by

$$
\mathrm{Cov}[\mathbf{x}_k, \mathrm{d}\mathbf{x}_k] = \mathbb{E}[\mathbf{x}_k\mathrm{d}\mathbf{x}_k] - \mathbb{E}[\mathbf{x}_k]\mathbb{E}[\mathrm{d}\mathbf{x}_k] = \mathbb{E}[\mathbf{x}_k\mathbf{f}(\mathbf{x}_k, \mathbf{u}_k)] - \boldsymbol{\mu}_k\mathrm{d}\boldsymbol{\mu}_k.
\tag{23}
$$

The kernel or hyper-parameter $\Theta = (\sigma_n, \sigma_s, \mathbf{W})$ can be learned by maximizing the log-likelihood of the training outputs given the inputs:

$$
\Theta^* = \underset{\Theta}{\mathrm{argmax}}\Big\{\log\Big(\mathrm{p}\big(\mathrm{d}\mathbf{X}|\tilde{\mathbf{X}}, \Theta\big)\Big)\Big\}.
\tag{24}
$$

where

$$
\begin{aligned}
\log\Big(\mathrm{p}\big(\mathrm{d}\mathbf{X}|\tilde{\mathbf{X}}, \Theta\big)\Big) = &-\frac{1}{2}\mathrm{d}\mathbf{X}^{\mathrm{T}}\Big(\mathbf{K}(\tilde{\mathbf{X}}, \tilde{\mathbf{X}}) + \sigma_n^2\mathbf{I}\Big)^{-1}\mathrm{d}\mathbf{X} \\
&-\frac{1}{2}\log\Big|\mathbf{K}(\tilde{\mathbf{X}}, \tilde{\mathbf{X}}) + \sigma_n^2\mathbf{I}\Big| - \frac{H}{2}\log 2\pi.
\end{aligned}
\tag{25}
$$

The optimization problem can be solved using numerical methods such as conjugate gradient.

## 3 Local dynamics models

In DDP related algorithms, a local model along a nominal trajectory $(\bar{\mathbf{x}}_k, \bar{\mathbf{u}}_k)$, where $k = 0, ..., H$, is created based on: i) a first or second-order local approximation of the dynamics model; ii) a second-order local approximation of the value function. In our proposed PDDP framework, we will create a local model along a trajectory of state distribution-control pair $(\mathrm{p}(\bar{\mathbf{x}}_k), \bar{\mathbf{u}}_k)$. We introduce the Gaussian augmented state vector $\mathbf{z}_k^x = [\boldsymbol{\mu}_k \ \mathrm{vec}(\boldsymbol{\Sigma}_k)]^{\mathrm{T}} \in \mathbb{R}^{n+n \times n}$ where $\mathrm{vec}(\boldsymbol{\Sigma}_k)$ is the vectorization of $\boldsymbol{\Sigma}_k$. First, we create a local linear model of the dynamics. Based on eq.(10), the dynamics model with the augmented state can be written as

$$
\mathbf{z}_{k+1}^x = \mathcal{F}(\mathbf{z}_k^x, \mathbf{u}_k).
\tag{26}
$$

Define the control and state variations $\delta\mathbf{z}_k^x = \mathbf{z}_k^x - \bar{\mathbf{z}}_k^x$ and $\delta\mathbf{u}_k = \mathbf{u}_k - \bar{\mathbf{u}}_k$. In this work we consider the first order expansion of the dynamics. More precisely we have

$$
\delta\mathbf{z}_{k+1}^x = \mathcal{F}_k^x\delta\mathbf{z}_k^x + \mathcal{F}_k^u\delta\mathbf{u}_k,
\tag{27}
$$

where the Jacobians $\mathcal{F}_k^x$ and $\mathcal{F}_k^u$ are specified as

$$\mathcal{F}_k^x = \nabla_{\mathbf{x}_k}\mathcal{F} = \left[ \begin{array}{cc} \frac{\partial \boldsymbol{\mu}_{k+1}}{\partial \boldsymbol{\mu}_k} & \frac{\partial \boldsymbol{\mu}_{k+1}}{\partial \boldsymbol{\Sigma}_k} \\ \frac{\partial \boldsymbol{\Sigma}_{k+1}}{\partial \boldsymbol{\mu}_k} & \frac{\boldsymbol{\Sigma}_{k+1}}{\partial \boldsymbol{\Sigma}_k} \end{array} \right] \in \mathbb{R}^{(n+n^2)\times(n+n^2)},$$

$$\mathcal{F}_k^u = \nabla_{\mathbf{u}_k}\mathcal{F} = \left[ \begin{array}{c} \frac{\partial \boldsymbol{\mu}_{k+1}}{\partial \mathbf{u}_k} \\ \frac{\partial \boldsymbol{\Sigma}_{k+1}}{\partial \mathbf{u}_k} \end{array} \right] \in \mathbb{R}^{(n+n^2)\times m}.$$

(28)

where the partial derivatives can be evaluated as

$$\begin{aligned} \frac{\partial \boldsymbol{\mu}_{k+1}}{\partial \boldsymbol{\mu}_k} &= \mathbf{I} + \frac{\partial \Psi^{\mathrm{T}} \mathbf{q}_k}{\partial \boldsymbol{\mu}_k} \\ &= \mathbf{I} + \frac{\partial \Psi^{\mathrm{T}} \mathbf{q}_k}{\partial \tilde{\boldsymbol{\mu}}_k} \frac{\partial \tilde{\boldsymbol{\mu}}_k}{\partial \boldsymbol{\mu}_k} \end{aligned}$$

(29)

For each output dimension, the partial derivative $\frac{\partial \Psi_i^{\mathrm{T}} \mathbf{q}_{ki}}{\partial \tilde{\boldsymbol{\mu}}_k} \frac{\partial \tilde{\boldsymbol{\mu}}_k}{\partial \boldsymbol{\mu}_k}$ can be obtain as

$$\begin{aligned} \frac{\partial \Psi_i^{\mathrm{T}} \mathbf{q}_{ki}}{\partial \tilde{\boldsymbol{\mu}}_k} \frac{\partial \tilde{\boldsymbol{\mu}}_k}{\partial \boldsymbol{\mu}_k} &= \sum_{j=1}^N \Psi_{ij} \frac{\partial \mathbf{q}_{kij}}{\partial \tilde{\boldsymbol{\mu}}_k} \frac{\partial \tilde{\boldsymbol{\mu}}_k}{\partial \boldsymbol{\mu}_k} \\ &= \left( \sum_{j=1}^N \Psi_{ij} \mathbf{q}_{kij} (\tilde{\mathbf{x}}_k - \tilde{\boldsymbol{\mu}}_k)^{\mathrm{T}} (\tilde{\boldsymbol{\Sigma}}_k + \mathbf{W}_j)^{-1} \right)^{\mathrm{T}} \frac{\partial \tilde{\boldsymbol{\mu}}_k}{\partial \boldsymbol{\mu}_k}. \end{aligned}$$

(30)

where $\frac{\partial \tilde{\boldsymbol{\mu}}_k}{\partial \boldsymbol{\mu}_k}$ can be easily obtained. Similarly, the partial derivatives of predictive mean with respect to state covariance for each output dimension can be found as

$$\begin{aligned} \frac{\partial \boldsymbol{\mu}_{k+1}}{\partial \boldsymbol{\Sigma}_k} &= \frac{\partial \Psi_i^{\mathrm{T}} \mathbf{q}_{ki}}{\partial \tilde{\boldsymbol{\Sigma}}_k} \frac{\partial \tilde{\boldsymbol{\Sigma}}_k}{\partial \boldsymbol{\Sigma}_k} = \sum_{j=1}^N \Psi_{ij} \frac{\partial \mathbf{q}_{kij}}{\partial \tilde{\boldsymbol{\Sigma}}_k} \frac{\partial \tilde{\boldsymbol{\Sigma}}_k}{\partial \boldsymbol{\Sigma}_k} \\ &= \sum_{j=1}^N \Psi_{ij} \mathbf{q}_{kij} \bigg( -\frac{1}{2} \Big( (\mathbf{W}_i^{-1}\tilde{\boldsymbol{\Sigma}}_k + \mathbf{I})^{-1} \mathbf{W}_i^{-1} \Big)^{\mathrm{T}} - \frac{1}{2} (\tilde{\mathbf{x}}_{ki} - \tilde{\boldsymbol{\mu}}_k)^{\mathrm{T}} \\ &\qquad \frac{\partial (\mathbf{W}_j + \tilde{\boldsymbol{\Sigma}}_k)^{-1}}{\partial \tilde{\boldsymbol{\Sigma}}_k} (\tilde{\mathbf{x}}_{ki} - \tilde{\boldsymbol{\mu}}_k) \bigg) \frac{\partial \tilde{\boldsymbol{\Sigma}}_k}{\partial \boldsymbol{\Sigma}_k}. \end{aligned}$$

(31)

where $\frac{\partial \tilde{\boldsymbol{\Sigma}}_k}{\partial \boldsymbol{\Sigma}_k}$ can be easily obtained. The partial derivatives of covariance with respect to input mean for each output dimension can be evaluated as

$$\frac{\partial \boldsymbol{\Sigma}_{(k+1)ij}}{\partial \boldsymbol{\mu}_k} = \left( \frac{\partial \mathrm{d}\boldsymbol{\Sigma}_{kij}}{\partial \tilde{\boldsymbol{\mu}}_k} + \frac{\partial \mathrm{Cov}[\mathbf{x}_{ki}, \mathrm{d}\mathbf{x}_{kj}]}{\partial \tilde{\boldsymbol{\mu}}_k} + \frac{\partial \mathrm{Cov}[\mathrm{d}\mathbf{x}_{ki}, \mathbf{x}_{kj}]}{\partial \tilde{\boldsymbol{\mu}}_k} \right) \frac{\partial \tilde{\boldsymbol{\mu}}_k}{\partial \boldsymbol{\mu}_k}$$

(32)

where

$$\begin{aligned} \frac{\partial \mathrm{d}\boldsymbol{\Sigma}_{kij}}{\partial \tilde{\boldsymbol{\mu}}_k} &= \Psi_i^{\mathrm{T}} \left( \frac{\partial \Phi_k}{\partial \tilde{\boldsymbol{\mu}}_k} - \frac{\partial \mathbf{q}_i}{\tilde{\boldsymbol{\mu}}_k} \mathbf{q}_j^{\mathrm{T}} - \mathbf{q}_i \frac{\partial \mathbf{q}_j}{\tilde{\boldsymbol{\mu}}_k} \right) \Psi_j + \left( -(\mathbf{K}^+ \sigma_n \mathbf{I})^{-1} \frac{\partial \Phi_k}{\partial \tilde{\boldsymbol{\mu}}_k} \right) \\ \frac{\partial \Phi_{kij}}{\partial \tilde{\boldsymbol{\mu}}_k} &= \Phi_{kij} \left( \frac{\mathbf{W}_j}{\mathbf{W}_i + \mathbf{W}_j} \tilde{\mathbf{x}}_i + \frac{\mathbf{W}_i}{\mathbf{W}_i + \mathbf{W}_j} \tilde{\mathbf{x}}_j - \tilde{\boldsymbol{\mu}}_k \right)^{\mathrm{T}} \left( \frac{1}{\mathbf{W}_i^{-1} + \mathbf{W}_j^{-1}} + \tilde{\boldsymbol{\Sigma}}_k \right)^{-1} \\ \frac{\partial \mathrm{Cov}[\mathrm{d}\mathbf{x}_k, \mathbf{x}_k]}{\partial \tilde{\boldsymbol{\mu}}_k} &= \frac{\tilde{\boldsymbol{\Sigma}}_k}{\tilde{\boldsymbol{\Sigma}} + \mathbf{W}} \sum_{i=0}^{n+m} \Psi \left( (\tilde{\mathbf{x}}_{ki} - \tilde{\boldsymbol{\mu}}_k) \frac{\partial \mathbf{q}_{ki}}{\partial \tilde{\boldsymbol{\mu}}_k} + \mathbf{q}_{ki} \mathbf{I} \right). \end{aligned}$$

(33)

The partial derivatives of covariance with respect to input covariance for each output dimension can be evaluated as

$$\frac{\partial \boldsymbol{\Sigma}_{(k+1)ij}}{\partial \boldsymbol{\Sigma}_k} = \mathbf{I} + \left( \frac{\partial \mathrm{d}\boldsymbol{\Sigma}_{kij}}{\partial \tilde{\boldsymbol{\Sigma}}} + \frac{\partial \mathrm{Cov}[\mathbf{x}_{ki}, \mathrm{d}\mathbf{x}_{kj}]}{\partial \tilde{\boldsymbol{\Sigma}}_k} + \frac{\partial \mathrm{Cov}[\mathrm{d}\mathbf{x}_{ki}, \mathbf{x}_{kj}]}{\partial \tilde{\boldsymbol{\Sigma}}_k} \right) \frac{\partial \tilde{\boldsymbol{\Sigma}}_k}{\partial \boldsymbol{\Sigma}_k}.$$

(34)

where

$$\frac{\partial d\mathbf{\Sigma}_{kij}}{\partial \tilde{\mathbf{\Sigma}}} = \Psi_i^{\mathrm{T}} \left( \frac{\partial \Phi_k}{\partial \mathbf{\Sigma}_k} - \frac{\partial \mathbf{q}_i}{\mathbf{\Sigma}_k} \mathbf{q}_j^{\mathrm{T}} - \mathbf{q}_i \frac{\partial \mathbf{q}_j}{\mathbf{\Sigma}_k} \right) \Psi_j + \left( -(\mathbf{K}^+ \sigma_n \mathbf{I})^{-1} \frac{\partial \Phi_k}{\partial \mathbf{\Sigma}_k} \right)$$

$$\frac{\partial \Phi_{kij}}{\partial \tilde{\mathbf{\Sigma}}_k} = -\frac{1}{2} \Phi_{kij} \Bigg[ \left( \left( \frac{\mathbf{W}_i + \mathbf{W}_j}{\mathbf{W}_i \mathbf{W}_j} \mathbf{\Sigma}_k + \mathbf{I} \right)^{-1} \left( \frac{\mathbf{W}_i + \mathbf{W}_j}{\mathbf{W}_i \mathbf{W}_j} \right) \right)^{\mathrm{T}}$$

$$- \left( \frac{\mathbf{W}_j}{\mathbf{W}_i + \mathbf{W}_j} \tilde{\mathbf{x}}_i + \frac{\mathbf{W}_i}{\mathbf{W}_i + \mathbf{W}_j} \tilde{\mathbf{x}}_j - \tilde{\boldsymbol{\mu}}_k \right)^{\mathrm{T}} \left( \frac{1}{\mathbf{W}_i^{-1} + \mathbf{W}_j^{-1}} + \mathbf{\Sigma}_k \right)^{-1}$$

$$\left( \frac{\mathbf{W}_j}{\mathbf{W}_i + \mathbf{W}_j} \tilde{\mathbf{x}}_i + \frac{\mathbf{W}_i}{\mathbf{W}_i + \mathbf{W}_j} \tilde{\mathbf{x}}_j - \tilde{\boldsymbol{\mu}}_k \right)$$

$$\frac{\partial \mathrm{Cov}[d\mathbf{x}_k, \mathbf{x}_k]}{\partial \tilde{\mathbf{\Sigma}}_k} = \left( \frac{1}{\tilde{\mathbf{\Sigma}}_k + \mathbf{W}} + \frac{\partial ((\tilde{\mathbf{\Sigma}} + \mathbf{W})^{-1})}{\partial \tilde{\mathbf{\Sigma}}_k} \right) \sum_{i=1}^{m+n} \Psi_{ki} \mathbf{q}_{ki} (\tilde{\mathbf{x}} - ti - \tilde{\boldsymbol{\mu}}_{ki}) +$$

$$\tilde{\mathbf{\Sigma}}_k \left( \frac{1}{\tilde{\mathbf{\Sigma}}_k + \mathbf{W}} \right) \sum_{i=1}^{n+m} \Psi_{ki} (\tilde{\mathbf{x}}_{ki} - \tilde{\boldsymbol{\mu}}_{ki}) \frac{\partial \mathbf{q}_{ki}}{\partial \tilde{\mathbf{\Sigma}}_k}. \tag{35}$$

We have found the expression of $\frac{\partial \boldsymbol{\mu}_{k+1}}{\partial \boldsymbol{\mu}_k}$, $\frac{\partial \boldsymbol{\mu}_{k+1}}{\partial \mathbf{\Sigma}_k}$, $\frac{\partial \mathbf{\Sigma}_{k+1}}{\partial \boldsymbol{\mu}_k}$, $\frac{\partial \mathbf{\Sigma}_{k+1}}{\partial \mathbf{\Sigma}_k}$ analytically. The partial derivatives with respective to control $\frac{\partial \boldsymbol{\mu}_{k+1}}{\partial \mathbf{u}_k}$, $\frac{\partial \mathbf{\Sigma}_{k+1}}{\partial \mathbf{u}_k}$ can be found similarly.

## 4  Cost function

In classic DDP/LQG and most optimal control problems, the following quadratic cost function was used:

$$\mathcal{L}(\mathbf{x}_k, \mathbf{u}_k) = (\mathbf{x}_k - \mathbf{x}_k^{goal})^{\mathrm{T}} \mathbf{Q} (\mathbf{x}_k - \mathbf{x}_k^{goal}) + \mathbf{u}_k^{\mathrm{T}} \mathbf{R} \mathbf{u}_k, \tag{36}$$

where $x_k^{goal}$ is the target state. In probabilistic DDP, given the distribution $\mathrm{p}(\mathbf{x}_k) \sim \mathcal{N}(\boldsymbol{\mu}_k, \mathbf{\Sigma}_k)$. Let $\sigma_{kij} = [\mathbf{\Sigma}_k]_{ij}$ and $q_{ij} = [\mathbf{Q}]_{ij}$. The expectation of original quadratic cost function can be obtained as:

$$\mathbb{E}\Big[ \mathcal{L}(\mathbf{x}_k, \mathbf{u}_k) \Big] = \mathbb{E}\Big[ (\mathbf{x}_k - \mathbf{x}_k^{goal})^{\mathrm{T}} \mathbf{Q} (\mathbf{x}_k - \mathbf{x}_k^{goal}) + \mathbf{u}_k^{\mathrm{T}} \mathbf{R} \mathbf{u}_k \Big]$$

$$= \mathbb{E}\Big[ \sum_{i=1}^{n} \sum_{j=1}^{n} q_{ij} (x_{ki} - x_i^{goal})(x_{kj} - x_j^{goal}) \Big] + \mathbf{u}_k^{\mathrm{T}} \mathbf{R} \mathbf{u}_k$$

$$= \sum_{i=1}^{n} \sum_{j=1}^{n} q_{ij} \mathbb{E}\Big[ (x_{ki} - x_i^{goal})(x_{kj} - x_j^{goal}) \Big] + \mathbf{u}_k^{\mathrm{T}} \mathbf{R} \mathbf{u}_k$$

$$= \sum_{i=1}^{n} \sum_{j=1}^{n} q_{ij} \bigg( \mathrm{Cov}\Big( (x_{ki} - x_i^{goal}), (x_{kj} - x_j^{goal}) \Big) +$$

$$\mathbb{E}\Big[ x_{ki} - x_i^{goal} \Big] \mathbb{E}\Big[ x_{kj} - x_j^{goal} \Big] \bigg) + \mathbf{u}_k^{\mathrm{T}} \mathbf{R} \mathbf{u}_k$$

$$= \sum_{i=1}^{n} \sum_{j=1}^{n} q_{ij} \Big( \sigma_{kij} + (\mu_{ki} - x_i^{goal})(\mu_{kj} - x_j^{goal}) \Big) + \mathbf{u}_k^{\mathrm{T}} \mathbf{R} \mathbf{u}_k$$

$$= \sum_{i=1}^{n} \sum_{j=1}^{n} q_{ij} \sigma_{tji} + \sum_{i=1}^{n} \sum_{j=1}^{n} q_{ij} (\mu_{ki} - x_i^{goal})(\mu_{kj} - x_j^{goal}) + \mathbf{u}_k^{\mathrm{T}} \mathbf{R} \mathbf{u}_k$$

$$= \sum_{i=1}^{n} [\mathbf{Q}\mathbf{\Sigma}_k]_{ii} + (\boldsymbol{\mu}_k - \mathbf{x}_k^{goal})^{\mathrm{T}} \mathbf{Q} (\boldsymbol{\mu}_k - \mathbf{x}_k^{goal}) + \mathbf{u}_k^{\mathrm{T}} \mathbf{R} \mathbf{u}_k$$

$$= \mathrm{tr}(\mathbf{Q}\mathbf{\Sigma}_k) + (\boldsymbol{\mu}_k - \mathbf{x}_k^{goal})^{\mathrm{T}} \mathbf{Q} (\boldsymbol{\mu}_k - \mathbf{x}_k^{goal}) + \mathbf{u}_k^{\mathrm{T}} \mathbf{R} \mathbf{u}_k$$

Therefore, in this paper we use the cost function with the augmented state

$$\mathcal{L}(\mathbf{z}_k^x, \mathbf{u}_k) = \mathrm{tr}(\mathbf{Q}\boldsymbol{\Sigma}_k) + (\boldsymbol{\mu}_k - \mathbf{x}_k^{goal})^{\mathrm{T}}\mathbf{Q}(\boldsymbol{\mu}_k - \mathbf{x}_k^{goal}) + (\mathbf{u}_k)^{\mathrm{T}}\mathbf{R}\mathbf{u}_k. \tag{37}$$

The partial derivatives of the above cost function with respect to $(\mathbf{z}_k^x, \mathbf{u}_k)$ can be easily obtained by

$$\frac{\partial}{\partial \mathbf{z}_k^x}\mathcal{L}(\mathbf{z}_k^x, \mathbf{u}_k) = \left[\frac{\partial}{\partial \boldsymbol{\mu}_k}\mathcal{L}(\mathbf{z}_k^x, \mathbf{u}_k) \quad \frac{\partial}{\partial \boldsymbol{\Sigma}_k}\mathcal{L}(\mathbf{z}_k^x, \mathbf{u}_k)\right]^{\mathrm{T}},$$

$$= \left[2(\boldsymbol{\mu}_k - \mathbf{x}^{goal})^{\mathrm{T}}\mathbf{Q} \quad \mathbf{Q}\right]^{\mathrm{T}},$$

$$\frac{\partial}{\partial \mathbf{u}_k}\mathcal{L}(\mathbf{z}_k^x, \mathbf{u}_k) = 2(\mathbf{u}_k)^{\mathrm{T}}\mathbf{R}.$$

The cost function scales linearly with the state covariance, therefore the exploration strategy of PDDP is balanced between the distance from the target and the variance of the state and avoids high risk explorations.