[Reviews · NeurIPS 2014]

Submitted by Assigned_Reviewer_1

The proposed approach, while straightforward, quite elegantly handles the problem at hand. What prevents this paper from being a clear cut acceptance is the lack of adequate experimental validation.

Typos line 47: draw -> drawn

A more thorough discussion of noise in the exploration step of Algorithm 1 (step 8) would be appreciated. This issue is also not discussed in the experiments section (how much noise was used?).

I also had a few issues with some of the claimed advantages in the paper. Specifically:
(1) The claim that PDDP has an advantage over PILCO since it does not have to solve non-convex optimization problems seems suspect given the non-convexity of the optimization problem solved in the hyper-parameter tuning step.

(2) The claim that PDDP’s complexity does not scale with the dimensionality of the state seems inaccurate. Even if it doesn’t factor into the computational complexity of the policy learning step, it clearly factors into the computation of the kernel matrices necessary for the GP inferences. This issue should be clarified in the final version.

(3) The claim on line 364 that PDDP keeps a fixed size of data seems only accurate in the special case where I_max = 1 (that is when trajectory collection and optimization are note iterated).

My biggest complaint with the paper is that the experimental results are not very strong. Firstly, there is no comparison of the total cost for PILCO versus PDDP. Secondly, is it the case that there are no straightforward methods to speedup PILCO? A discussion of this point would be appreciated. Lastly, GPDDP appears to do as well or better than PDDP on every dimension (data efficiency and computational efficiency). This seems to undermine the idea that PDDP has the benefit of doing "safe exploration". A discussion of this point would strengthen the experiments section.
Summary: The authors present a very nice, approach to combining Gaussian Process Regression and Differential Dynamic Programming that elegantly handles the exploration / exploitation tradeoff. The principal downside of the paper is in the experimental validation.

Submitted by Assigned_Reviewer_22

This paper proposes a DDP formulation for models represented as Gaussian processes.
The derivations of the required equations that connects the Gaussian Process model with the DDP framework are provided.
The method allows the solution of local optimal trajectory problems for systems whose dynamics are not known in advance.
As such this work shares several characteristics with PILCO. The simulated results show that solutions are found much faster than PILCO.

Quality and clarity

Although the concept of the paper is strong, the paper needs more work to be done, particularly regarding the presentation of the results.
Presentation of the results is currently too shallow, especially the discussion, which is basically non-existent for figures 1 and 2.
Claims such as "safe exploration" lacks support and the computational complexity conclusion is questionable.
The writing could be refined in general; I am not referring to specific sentences, but it feels this work is lacking some additional number of polishing iterations on the writing.

Originality and Significance

The fact that DDP is being proposed as the optimization method has an impact for the DDP/iLQG practitioners and also for the model-based optimization community. Unarguably the presented work shares several similarities with PILCO both in the philosophy and also the methodology. However, while the work of PILCO basically kept the choices of the optimizer open, this work suggest DDP and develops the full framework thus providing a self-contained solution.
This work can be much improved if extensive and careful analyses between PDDP and PILCO are provided.

Questions

. Line 33: "... Compared to global optimal control approaches, the local optimal DDP shows superior scalability to high-dimensional problems..." is an incomplete statement. The big trade-off is that DDP solution is only local and prone to provide a poor a local optima. I agree that for high-dimensional problems a local solution is usually the only option. If there was no trade-off there would be no reason for the full dynamic programming.
. Line 212: The sentence " ... where the variance of control distribution and the off-diagonal entries of the state covariance matrix can be neglected because of no impact on control performances." is not obvious especially regarding the control distribution. Noise in control can have a large impact on how the states integrate over time. It may also influence the final cost. Are you assuming a deterministic system?
. Line: 294. The conclusion on the computational complexity is precipitated. Yes, although the complexity of the learning policy does not relate directly to the number of states, for the usual case of fully actuated systems, the number of control inputs is proportinal to the number of states which makes the complexity of the algorithm O3.
. The formulation of the "safe exploration" feature is not clear in the text. Although Figure 3(a) is a cartoon that shows the intuition, where is this intuition implemented in the algorithm?
. Line 322: "To keep the system stable" seems to be a bad sentence, unless you are talking about unstable system such as an inverted pendulum. Do you mean "final zero velocity"?
. The variance in (a) seems simply to small, almost a deterministic system. As the GP is learning the system from samples one can infer that the variance during the initial iterations are large and decrease with the number iterations. It would be interesting to see such a plot.
. I do not see the point of proposing another method (GPDDP) that does not bring any benefit. What is the motivation for that? If what you want to show is the effect of the propagation of uncertainty then you can just say that without the need of introducing the GPDDP as a new second method.
. The paper must also compare the traditional DDP results when dynamics are fully known in Figs 1 and 2.

Minor questions:
. Legends for Figure 1 are too small. The y axes of (a) and (c) should be on the same range.
. Why not showing the costs for PILCO on figures 1(b) and 2(b)?
Summary: The paper provides a self-contained framework for GP-based DDP which is of high interest for the DDP/iLQG learning community. The paper, however, is not clear enough and the evaluation, particularly in regards to PILCO, is insufficient.

Submitted by Assigned_Reviewer_35

The authors propose to use a Gaussian Process (GP) dynamics model learned from data together with a DDP-like trajectory optimization algorithm operating on the mean and covariance of the state distribution. The proposed method is compared to PILCO and an ablated variant (using just the mean as the state), and achieves slightly worse results but at a very large improvement in computation time.

It appears that, besides learning a GP model of the dynamics, the proposed method is simply performing Gaussian belief space planning. This connection is not mentioned by the authors, but seems pretty clear. For example, the paper "Motion planning under uncertainty using iterative local optimization in belief space" describes a method that, except for the GP learning phase, appears extremely similar (there are numerous other methods in this general "family" of belief space planning techniques). The novelty of the approach therefore would seem to be quite low, as the method is simply taking belief space planning and combining it with learned GP models. Since belief space planning algorithms are typically agnostic to the form of the simulator used, a GP-based dynamics model would be straightforward to combine with any previous belief space planning algorithm, yielding an extremely similar approach.

There are several other issues with this work. Among previous methods, the authors only compare to PILCO. The body of belief space planning work is completely ignored, though perhaps this is because such work often does not consider learning the model (since any model learning technique can easily be used in conjunction with any belief space planner). However, there is also other work on model-based RL to compare to. Since the authors are claiming computation time as a major advantage of their method, the choice of PILCO doesn't really make sense, since PILCO has exceedingly extravagant computational requirements that are far outside the "norm" for model-based RL. Without a comparison to any other method, it's not clear whether the improvement in the tradeoff between computation and sample complexity is actually notable.

The title of the paper is also a bit too ambitious. There are plenty of papers that discuss the "probabilistic" aspects of DDP, including a paper entitled "Stochastic DDP," papers that deal with noise and uncertainty (including state and action dependent noise, etc), and of course the previously mentioned field of belief space planning. Something like Gaussian Process DDP would have been more appropriate, but since the primary contribution of the paper appears to be combining belief space planning with learned GP models, something about "GP models for belief space planning" might also make sense.

-----

Regarding the rebuttal: I'm not entirely clear on the distinction that the authors are drawing between their approach and belief-space planning. In belief-space planning, the underlying *true* dynamics is sometimes assumed to be Gaussian (not the same as deterministic, but it's true that E[f(x)] = f(E[x])). But the resulting mean and covariance propagation equations look similar to the ones presented in this paper: stochastic transitions in terms of the state are turned into deterministic transitions in terms of the state mean and covariance. If there is some distinction between them, it should be demonstrated experimentally, as it is not at all obvious from the description whether one method is better or worse.

I do appreciate the clarification with PI^2, though I'm not sure if this algorithm is the best candidate for a comparison, since requiring 2500 iterations to solve pendulum swingup seems quite excessive (see for ex NP-ALP, Reinforcement Learning In Continuous Time and Space Doya '00, and other papers that compare on the pendulum benchmark). Another *model-based* method would be a better candidate here I think, since you are comparing model-based methods with a particular kind of inductive bias (smooth dynamics functions).
Summary: The proposed method appears to be DDP-based belief space planning combined with a learned GP dynamics model. In light of prior work, this doesn't seem particularly novel. The experimental results also fail to convincingly demonstrate the advantage of the proposed method in terms of the computation cost-sample complexity tradeoff, since the only prior method compared to (PILCO) has unusually high computational cost compared to most model-based RL methods.

Submitted by Meta_Reviewer_4

The results presented are really strong.

Section 2.1 is pretty much what is written in [5]. A reference along
the lines "taken from [5]" or "along the lines of the derivations in
[5]" would be appropriate. This would also make room to shorten this
part and expand on Sectoin 2.2 and/or the experimental results.

I have a few questions about section 2.2.1:
- - Why are we looking only at p(x) and p(u) and not p(x,u)? This joint
can be computed (see section 2.1).

Line 121: It's not p(dx_t|\tilde x_t) that is computed but the mean
and covariance of p(dx_t). The conditional would be a standard GP
prediction and does not require approximations. The marginal p(dx_t)
however will require approximations. In this paper, the authors do
moment matching. It shoule be mentioned that the mean/covariance can
be computed exactly, but in the end we have a Gaussian approximation
to the predictive distribution.

Eq (12) is missing a bracket

Eq 17: With the augmented variable z = [mu \Sigma] and z_{t+1} =
F(z_t) can we still guarantee that z_{t+1} contains a valid covariance
(symmetric, positive definite)? It would be good to discuss this.

The authors say that they consider linear controllers. If they are
linear state-feedback controllers of the form u = Cx then it seems
that the gradients in Eq 19 might be incomplete. If we only look at
the mean of z_t, I would compute the gradients as

dmu_{t+1}/dmu_t^x
= \partial mu_{t+1}/\partial mu_t^x
+ \partial mu_{t+1}/\partial mu_t^u * \partial mu_t^u/\partial mu_t^x
+ \partial mu_{t+1}/\partial Sigma_t^u * \partial Sigma_t^u/\partial
mu_t^x

The last two terms don't appear. The very last term might be 0 because
of the linearity of the controller, but I don't see why the second
term doesn't show up. Similar arguments would hold for other gradients
as well.

In the cost function (Eq. 21 and above): Why do we not take the
control distribution into account properly. I believe this would then
result in
E[L(x,u)] = tr(Q\Sigma_t) + (mu_t - x_t^goal)^T Q (mu_t - x_t^goal) +
\mu_t^T * R * \mu_t^u + tr(Sigma_t^u * R)

The last term, which involves the covariance of the control signal, is
missing. Why is that?

Line 211: Why can we ignore the variance of the control signal? If I
look at Eq. 21 I would agree, but since I believe that equation 21
should depend on the variance of the control signal as well I cannot
follow the statement in line 211.

Section 2.4: How does the computational complexity depend on the state
dimension.

Experimental results:

I'm impressed by the results presented. However, I have a couple of
comments:

- - I'm not convinced at all that N<=3 trajectories are enough to build
a half-way decent GP dynamics model. Even if these trajectories are
sampled with the optimal policy it will be very hard, especially for
very nonlinear systems such as the CDIP. Please explain why so little
data is sufficient to get a reasonable model.

- - Details about the experimental results are missing largely: How long
was the trial, what was evaluated, how were the critical parameters set?

- - Did you use the same cost function for GPDDP and PILCO? Or did you
use different cost functions?

- - How was Fig. 1c generated? If I look at the video
https://www.youtube.com/watch?v=ki9BZeugVxc&noredirect=1 it seems that
the PILCO algorithm can stabilize the double pendulum after a few
seconds (once a good policy is found). This might depend on different
parameter settings, but it should be explained.

- - Figure 4 (a) reports (I believe) not the best case performances; at
least not for PILCO. According to [3], PILCO requires in a "typical
experiment" 20-30 trials.

- - Figure 4 (b) reports the "total time consumed to obtain the final
policy". This means the computation time? If so, please clarify.

- - Line 368: "The saturating cost defined in PILCO leads to more
explorations in high cost regions" is not necessarily correct.
According to the discussion in [5] it leads to exploration in the
early stages of learning and to exploitation in the late stages of
learning.
Summary: Interesting paper, leaves some confusion.
Author Feedback
Author rebuttal: We thank all reviewers for their very constructive comments. We will correct typos and improve the manuscript in terms of grammatical errors and presentation style as suggested by the reviewers. A supplementary material of this feedback form can be found at
https://www.dropbox.com/s/him299gf6eq5taj/rebuttal_sup.pdf
which includes some of the plots requested by the reviewers and equations/references to support our responses.

To reviewer_1:
Experimental results:
We did not show the cost for PILCO because it uses a different cost function rather than quadratic cost we used in the manuscript (as in many RL works), therefore cost comparison is less intuitive. However, we provide the cost comparison in the supplementary material (above link).
In addition, Fig. 4 (a) shows that GPDDP requires more interactions with the physical system than PDDP, therefore PDDP performs better in terms of data-efficiency.
Noises: we applied zero-mean normally distributed noise (function 'randn' in Matlab) to the control.
Other issues:
(1) What we mean here is that PDDP does not rely on any optimizer for learning policies. However, we do use nonconvex optimization tools to learn the hyper parameter of the GP as in other GP-based approaches. We will rephrase this point to avoid confusions.
(2) We agree with the reviewer's point, we will clarify this point in the revised version.
(3) When necessary, PDDP updates its training set by interacting with the physical systems (sample data), the data from previous iterations would be removed, therefore PDDP keeps a fixed size of training data.

To reviewer_22:
We will improve the presentation especially for the results analysis and rephrase some of the claims to avoid confusion.
Line 33: We agree that there is a trade-off for PDDP since it is based on local optimization. We will make this point clear.
Line 212: What we mean here is that the cost function does not incorporate the control variance. We will rephrase this point.
Line 294 and 322: We will rephrase/clarify these statements.
'Safe exploration': We have implemented this scheme in all examples. Fig.3 (b) shows PDDP learning scheme would stay 'closer' to the sampled data than GPDDP (without this feature). We will make this point clear.
Small variances of Fig. 1(a): As we have shown in the manuscript, whenever PDDP updates GP training set by interacting with physical systems (Algorithm step 8), the variances of state predictions would reduce significantly. (a) shows the final optimized trajectories, an interaction during late learning phase would cause relatively small predictive variances.
Regarding GPDDP: GPDDP controls the expected state rather than the state distribution, therefore it does not have the same exploration scheme as PDDP. Comparison of PDDP and GPDDP shows the effect of exploration scheme that lead to better data efficiency. However, we will consider using a different method for comparison.
Comparison with traditional DDP: please refer to the supplementary (link above) for this comparison.
Not showing PILCO cost: PILCO uses a different cost function, therefore the comparison seems less intuitive.

To reviewer_35:
With respect to the similarity of the Gaussian belief space planning frameworks with the manuscript: we are fully aware of this family of works. The paper mentioned by the reviewer is an important piece and we will mention the connections of the aforementioned paper with our work in the revised version. However, PDDP has some significant differences compared to the works suggested by the reviewer:
PDDP incorporates general uncertainty (not only the uncertainty due to process noise). Even if we ignore all “learning dynamics” parts from the manuscript, that is, if we assume that a dynamical system is given with Gaussian state distribution, the state propagation in PDDP is still very different from [3][4] (please see the supplementary for references/details). For instance, equations 14-17 of [4] indicate the state x is stochastic but the state transition function f is deterministic. In PDDP, both x and f are stochastic. This results in very different belief dynamics, especially when propagating state distributions over a long time horizon, this difference in belief dynamics affects the final polices significantly.

In many belief space-planning frameworks (e.g. [3][4]), a typical formulation is for a controller to control the Kalman filter (EKF, etc). This is a valid formulation for as long as the filter is consistent and stable. Consistency is true for as long the error (residual) between the true and predictive measurement and/or state is within the 3 sigma bounds of the corresponding covariance. Lack of consistency results in divergence of KF, which means that the KF can no longer represent the true dynamics (the estimated state is far from the true state). Applying optimal control to a diverging filter leads to undesirable results. PDDP deals with the issue of a belief representation that is far from the true dynamics by allowing interactions with the real physical system.

Comparison: We agree that PILCO is not a fast RL framework, but it is probably the most data-efficient one. As requested by the reviewer, we are comparing PDDP with a faster RL method: the path integral approach (PI). PI computes the optimal policy based on sampled trajectories, and does not require any gradient-based solver. The results can be found in the supplementary material. We are not claiming PDDP to be the fastest in terms of policy learning, however we believe that the major advantage of PDDP is the combination of data-efficiency and policy learning speed.

Title: We will re-consider the title. The “stochastic DDP” mentioned by the reviewer is a generalization of the classic DDP to stochastic dynamics, while PDDP deals with learning policy from data.